# Dual Feature Reduction for the Sparse-group Lasso and its Adaptive Variant

**Fabio Feser** [1]   **Marina Evangelou** [1]

## Abstract

The sparse-group lasso performs both variable and group selection, simultaneously using the strengths of the lasso and group lasso. It has found widespread use in genetics, a field that regularly involves the analysis of high-dimensional data, due to its sparse-group penalty, which allows it to utilize grouping information. However, the sparse-group lasso can be computationally expensive, due to the added shrinkage complexity, and its additional hyperparameter that needs tuning. This paper presents a novel feature reduction method, Dual Feature Reduction (DFR), that uses strong screening rules for the sparse-group lasso and the adaptive sparse-group lasso to reduce their input space before optimization, without affecting solution optimality. DFR applies two layers of screening through the application of dual norms and subdifferentials. Through synthetic and real data studies, it is shown that DFR drastically reduces the computational cost under many different scenarios.

## 1. Introduction

High-dimensional datasets, where the number of features ($p$) is far greater than the number of observations ($n$) in a matrix $\mathbf{X} \in \mathbb{R}^{n \times p}$, are becoming increasingly common with the increased rate of data collection. To handle this, *shrinkage methods*, such as the lasso (Tibshirani, 1996), elastic-net (Zou & Hastie, 2005), and SLOPE (Bogdan et al., 2015) have been proposed and found increased use in the machine learning community (Alaoui & Mahoney, 2015; Michoel, 2018; Lemhadri et al., 2021; Thompson et al., 2023). These methods shrink estimates towards zero during optimization, enabling *variable selection*, to identify which features, $\beta \in \mathbb{R}^p$, have an association with the response $y \in \mathbb{R}^n$.

[1]Department of Mathematics, Imperial College London, London, UK. Correspondence to: Fabio Feser <ff120@ic.ac.uk>.

*Proceedings of the 42^{nd} International Conference on Machine Learning*, Vancouver, Canada. PMLR 267, 2025. Copyright 2025 by the author(s).

In genetics, these methods help identify genes associated with disease outcomes. As genes are naturally found in groups (pathways), *group selection* approaches have been proposed, that allow grouping information to be used, such as the group lasso (Yuan & Lin, 2006), group SLOPE (Brzyski et al., 2019), and group SCAD (Guo et al., 2015). Applying only group shrinkage can harm convergence and prediction, as all variables in an active group are retained, including noise variables (Simon et al., 2013; Feser & Evangelou, 2023).

This limitation led to the development of sparse-group models, such as the Sparse-group Lasso (SGL) (Simon et al., 2013) and Sparse-group SLOPE (SGS) (Feser & Evangelou, 2023). These models can shrink variables in active groups by applying shrinkage on both variables and groups to yield bi-level selection. SGL has found increased popularity in applications in the machine learning (Vidyasagar, 2014; Yogatama & Smith, 2014) and healthcare (Peng et al., 2010; Simon et al., 2013; Fang et al., 2015) communities. Sparse-group models have been shown to have tangible benefits by consistently outperforming the lasso and group lasso in selection and prediction tasks (Simon et al., 2013; Feser & Evangelou, 2023).

Suppose the variables sit in a grouping structure, with disjoint sets of variables $\mathcal{G}_1, \ldots, \mathcal{G}_m$ of sizes $p_1, \ldots, p_m$. Then, SGL is a convex combination of the lasso and group lasso (Simon et al., 2013):

$$\hat{\beta}_{\text{sgl}}(\lambda) \in \underset{\beta \in \mathbb{R}^p}{\arg \min} \{f(\beta) + \lambda \|\beta\|_{\text{sgl}}\}, \tag{1}$$

$$\text{where } \|\beta\|_{\text{sgl}} = \alpha \|\beta\|_1 + (1-\alpha) \sum_{g=1}^{m} \sqrt{p_g} \|\beta^{(g)}\|_2, \tag{2}$$

such that $f$ is a differentiable and convex loss function, $\lambda > 0$ defines the level of shrinkage, $\beta^{(g)} \in \mathbb{R}^{p_g}$ is the vector of coefficients in group $g$, and $\alpha \in [0, 1]$. SGL has been extended to have adaptive shrinkage through the adaptive sparse-group lasso (aSGL) (Poignard, 2020; Mendez-Civieta et al., 2021) (Section 3.3).

### 1.1. Feature Reduction Approaches for the Sparse-group Lasso

The strengths of SGL come with increased computational cost, due to the additional shrinkage and the tuning of two

hyperparameters. Typically, $\alpha$ is set subjectively (Simon et al. (2013) suggest $\alpha = 0.95$) and $\lambda$ is tuned using cross-validation along a path $\lambda_1 \geq \ldots \geq \lambda_l \geq 0$. Algorithms such as the Least Angle Regression (LARS) approach calculate solutions for all possible values of $\lambda$, but are very sensitive to multicollinearity and scale quadratically, rendering their use in high-dimensional settings limited (Efron et al., 2004). Instead, feature reduction techniques, including screening rules, can help ease the cost of fitting a model along a path by discarding features before optimization that would have been inactive at the optimal solution. Whilst methods exist to discard observations (Shibagaki et al., 2016; Zhang et al., 2017), the focus of this paper is high-dimensional settings, in which discarding features is more impactful on computational savings.

Feature reduction techniques are either *exact* or *heuristic*. Exact methods strictly discard only inactive features but are conservative, while heuristic methods discard more features at the risk of violations. These violations are countered by checking the Karush–Kuhn–Tucker (KKT) conditions (Kuhn & Tucker, 1950) and adding any offending features back into the optimization. Heuristic rules discard significantly more variables than exact rules, providing large computational savings (Tibshirani et al., 2010).

Most exact methods follow the seminal Safe Feature Elimination (SAFE) framework (El Ghaoui et al., 2010), which has been applied to the group lasso (Bonnefoy et al., 2015) and SGL (Ndiaye et al., 2016a). Other exact examples include the dome test (Xiang & Ramadge, 2012), Dual Polytope Projections (DPP) (Wang et al., 2013), and Slores (Wang et al., 2014). The strong rule by Tibshirani et al. (2010) provides a framework for applying heuristic reduction with single separable norms, which has been extended to non-separable (Larsson et al., 2020) and sparse-group norms (Feser & Evangelou, 2025). Other examples include Sure Independence Screening (SIS) (Fan & Lv, 2008) and the Hessian rule (Larsson & Wallin, 2024).

Aside from the exact and heuristic categories, feature reduction techniques tend to follow three forms: *static*, where the feature reduction occurs only once at the start (El Ghaoui et al., 2010; Xiang et al., 2011; Xiang & Ramadge, 2012), *dynamic*, where reduction occurs iteratively (Bonnefoy et al., 2015), and *sequential*, where information from the previous solution is used (Tibshirani et al., 2010; Larsson et al., 2020; Larsson & Wallin, 2024; Feser & Evangelou, 2025).

An exact reduction method for SGL, called GAP safe, was proposed by Ndiaye et al. (2016a) using the SAFE framework. GAP safe uses the duality gap to create feasible regions where the active variables sit and applies reduction on the groups and variables. Other reduction methods for SGL include Two-layer Feature Reduction (TLFre) (exact) (Wang & Ye, 2014), though it was shown not to be exact

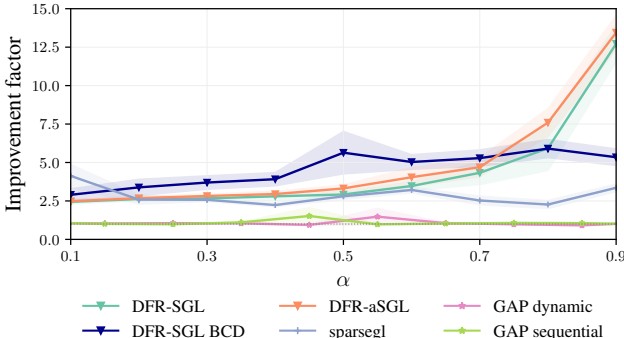

Figure 1: The improvement factor (high is best), which measures by how many orders the screening has improved fitting time, shown for different SGL screening methods, as a function of $\alpha$, with $95\%$ confidence intervals. The data was generated under a linear model with even groups of sizes 20 (Section 4.1). The two GAP methods have very similar values so jitter was added.

(Ndiaye et al., 2015), and sparsegl (heuristic) (Liang et al., 2022), which applies only group-level reduction. Additional speed-up attempts include using approximate bounds for inactive conditions (Ida et al., 2019) and a heuristic screening rule limited to multi-response Cox modeling (Li et al., 2022).

## 1.2. Contributions

In this paper, we propose a new feature reduction method for SGL and adaptive SGL, *Dual Feature Reduction* (DFR), which is based on the strong rule (Tibshirani et al., 2010) and the sparse-group screening framework (Feser & Evangelou, 2025). DFR introduces the first bi-level strong screening rules for SGL and the first screening rules for aSGL.

DFR applies two layers of screening, discarding inactive groups and inactive variables within active groups. By reducing the input dimensionality before optimization, DFR enables expanded tuning regimes to be performed, including concurrent tuning of $\lambda$ and $\alpha$. These benefits are achieved without affecting solution optimality. The computational efficiency of DFR increases the accessibility of SGL and aSGL models, encouraging wider adoption across fields.

The GAP safe rules for SGL require computation of safe regions, which includes a radius and center, and the dual norm, as well as iterative screening and fitting. In contrast, DFR needs only the dual norm and screens only once at each path point, making it considerably less expensive, as evidenced by our results (Figure 1).

DFR is described for SGL in Section 3 and then extended to aSGL in Section 3.3. The proofs of the results presented in these sections are provided in Appendix A.1 for SGL and

Appendix B.2 for aSGL. DFR is applied to synthetic and real data in Sections 4 and 5, where it is found to be the state-of-the-art screening approach for SGL, outperforming other existing methods, while still achieving the optimal solution.

## 2. Preliminaries

### 2.1. Problem Statement

SGL is trained along a path of parameters $\lambda_1 \geq \ldots \geq \lambda_l \geq 0$. The objective is to use the solution at $\lambda_k$ to generate a set of *candidate variables* $\mathcal{C}_v(\lambda_{k+1}) \subset [p] := \{1, \ldots, p\}$, that is a superset of the (unknown) set of active variables at $\lambda_{k+1}$, given by $\mathcal{A}_v(\lambda_{k+1}) := \{i \in [p] : \hat{\beta}_i(\lambda_{k+1}) \neq 0\}$. The optimization at $\lambda_{k+1}$ (Equation 1) is then calculated using only $\mathcal{C}_v(\lambda_{k+1})$. If the candidate set is a small proportion of the total input space, large computational savings are expected.

To generate the candidate variable set, we first generate a candidate group set (Section 3.1.1), which is then used as a basis for constructing the candidate variable set (Section 3.1.2). This is done using the dual norm of SGL.

### 2.2. Dual Norm

DFR requires evaluating the dual norm of SGL, defined as $\|z\|_{\mathrm{sgl}}^* := \sup\{z^\top x : \|x\|_{\mathrm{sgl}} \leq 1\}$. The SGL norm can be expressed in terms of the dual of the $\epsilon$-norm (Ndiaye et al., 2016a),

$$\|\beta\|_{\mathrm{sgl}} = \sum_{g=1}^{m} \tau_g \|\beta^{(g)}\|_{\epsilon_g}^*, \tag{3}$$

where $\tau_g = \alpha + (1-\alpha)\sqrt{p_g}$. The $\epsilon$-norm, $\|x\|_{\epsilon_g}$, applied to a group $g \in [m]$, is defined as the unique nonnegative solution $q$ of the equation (Burdakov, 1988)

$$\sum_{i=1}^{p_g} (|x_i| - (1-\epsilon_g)q)_+^2 = (\epsilon_g q)^2, \quad \epsilon_g = \frac{\tau_g - \alpha}{\tau_g}. \tag{4}$$

Using this, by Ndiaye et al. (2016a), the dual norm of SGL applied to a group $g \in [m]$ can be formulated as

$$\|\xi^{(g)}\|_{\mathrm{sgl}}^* = \max_{g=1,\ldots,m} \tau_g^{-1} \|\xi^{(g)}\|_{\epsilon_g}. \tag{5}$$

## 3. Dual Feature Reduction

DFR is first derived for SGL (Section 3.1) and then extended to aSGL (Section 3.3). DFR is summarised for SGL and aSGL in Table A1.

### 3.1. Sparse-group Lasso

#### 3.1.1. GROUP REDUCTION

To generate a candidate group set, the KKT stationarity conditions (Kuhn & Tucker, 1950) are used, providing conditions for an inactive group. For SGL, they are given by, for a group $g \in [m]$ at $\lambda_{k+1}$ (using the $\epsilon$-norm representation in Equation 3)

$$\mathbf{0} \in \nabla_g f(\hat{\beta}(\lambda_{k+1})) + \tau_g \lambda_{k+1} \Theta_{g,k+1}, \tag{6}$$

where $\Theta_{g,k+1} = \partial \|\hat{\beta}(\lambda_{k+1})\|_{\epsilon_g}^*$ is the subgradient of the dual norm of the $\epsilon$-norm at $\lambda_{k+1}$. The subgradient for an inactive group $g$ (at zero) can be expressed by the unit ball of the dual norm (Schneider & Tardivel, 2022):

$$\Theta_{g,k+1}^0 := \partial \|0\|_{\epsilon_g}^* = \left\{ x \in \mathbb{R}^{p_g} : \|x\|_{\epsilon_g} \leq 1 \right\}.$$

Plugging the unit ball into Equation 6 and applying the $\epsilon$-norm, the subgradient can be canceled out, so the KKT conditions can be written as

$$\|\nabla_g f(\hat{\beta}(\lambda_{k+1}))\|_{\epsilon_g} = \tau_g \lambda_{k+1} \|\Theta_{g,k+1}^0\|_{\epsilon_g} \leq \tau_g \lambda_{k+1}. \tag{7}$$

If the gradient at $\lambda_{k+1}$ were available, we could exactly identify the active groups (Proposition 3.1).

**Proposition 3.1** (Theoretical SGL group screening). *For SGL applied with any $\lambda_{k+1}, k \in [l-1]$, the candidate group set,*

$$\mathcal{C}_g(\lambda_{k+1}) = \{g \in [m] : \|\nabla_g f(\hat{\beta}(\lambda_{k+1}))\|_{\epsilon_g} > \tau_g \lambda_{k+1}\},$$

*is such that $\mathcal{C}_g(\lambda_{k+1}) = \mathcal{A}_g(\lambda_{k+1}) := \{g \in [m] : \|\hat{\beta}^{(g)}(\lambda_{k+1})\|_2 \neq 0\}$.*

However, as this is not possible in practice, an approximation $\mathcal{M}_g$ is required such that

$$\|\nabla_g f(\hat{\beta}(\lambda_{k+1}))\|_{\epsilon_g} \leq \mathcal{M}_g. \tag{8}$$

Then, the screening rule tests whether $\mathcal{M}_g \leq \tau_g \lambda_{k+1}$. If this is found to be true, it can be concluded that Equation 7 holds and the group must be inactive. An approximation can be found by assuming that the gradient is a Lipschitz function of $\lambda_{k+1}$ with respect to the $\epsilon$-norm,

$$\|\nabla_g f(\hat{\beta}(\lambda_{k+1})) - \nabla_g f(\hat{\beta}(\lambda_k))\|_{\epsilon_g} \leq \tau_g |\lambda_{k+1} - \lambda_k|, \tag{9}$$

which is a similar assumption to that used in the lasso strong rule (Tibshirani et al., 2010). Using the reverse triangle inequality gives

$$\|\nabla_g f(\hat{\beta}(\lambda_{k+1}))\|_{\epsilon_g} \leq \underbrace{\|\nabla_g f(\hat{\beta}(\lambda_k))\|_{\epsilon_g} + \tau_g(\lambda_k - \lambda_{k+1})}_{=:\mathcal{M}_g},$$

yielding a suitable approximation $\mathcal{M}_g$. Therefore, the strong group screening rule for SGL (Proposition 3.2) can

be formulated by plugging $\mathcal{M}_g$ into Equation 8: discard a group $g \in [m]$ if

$$\|\nabla_g f(\hat{\beta}(\lambda_k))\|_{\epsilon_g} \leq \tau_g(2\lambda_{k+1} - \lambda_k). \quad (10)$$

Since the Lipschitz assumption can fail, KKT checks (Section 3.1.3) are performed to prevent violations.

**Proposition 3.2** (DFR-SGL group screening). *For SGL applied with any $\lambda_{k+1}, k \in [l-1]$, assuming that*

$$\|\nabla_g f(\hat{\beta}(\lambda_{k+1})) - \nabla_g f(\hat{\beta}(\lambda_k))\|_{\epsilon_g} \leq \tau_g |\lambda_{k+1} - \lambda_k|,$$

*for all $g \in [m]$, then the candidate group set,*

$$\mathcal{C}_g(\lambda_{k+1}) = \{g \in [m] :$$
$$\|\nabla_g f(\hat{\beta}(\lambda_k))\|_{\epsilon_g} > \tau_g(2\lambda_{k+1} - \lambda_k)\},$$

*is such that $\mathcal{A}_g(\lambda_{k+1}) \subset \mathcal{C}_g(\lambda_{k+1})$.*

### 3.1.2. VARIABLE REDUCTION

Group screening reduces the input dimensionality, but further reduction is possible by applying a second screening layer to the variables in the candidate groups. For an inactive variable, $i \notin \mathcal{A}_v(\lambda_{k+1}), i \in \mathcal{G}_g$, the KKT conditions are (by Equation 1)

$$\mathbf{0} \in \nabla_i f(\hat{\beta}(\lambda_{k+1})) + \lambda_{k+1}\alpha\Phi_{i,k+1}^0 + \lambda_{k+1}(1-\alpha)\Psi_{i,k+1}^{(g)}, \quad (11)$$

where $\Phi$ and $\Psi$ are the subgradients of the $\ell_1$ and $\ell_2$ norms respectively. For an active group, the subgradient of the $\ell_2$ norm is given by $\hat{\beta}_i^{(g)}/\|\hat{\beta}^{(g)}\|_2$, which vanishes for an inactive variable. So,

$$-\nabla_i f(\hat{\beta}(\lambda_{k+1})) \in \lambda_{k+1}\alpha\Phi_{i,k+1}^0$$
$$\iff |\nabla_i f(\hat{\beta}(\lambda_{k+1}))| \leq \lambda_{k+1}\alpha, \quad (12)$$

where $\Phi_{i,k+1}^0 = \{x \in \mathbb{R} : |x| \leq 1\}$. As before, knowledge of the gradient would lead to exact support recovery (Proposition A.1). Equation 12 is similar to the strong screening rule for the lasso (Tibshirani et al., 2010), scaled by $\alpha$. Hence, using a scaled version of Lipschitz assumption for the lasso, the variable screening rule for SGL is formulated in Proposition 3.3.

**Proposition 3.3** (DFR-SGL variable screening). *For SGL applied with any $\lambda_{k+1}, k \in [l-1]$, assuming that*

$$|\nabla_i f(\hat{\beta}(\lambda_{k+1})) - \nabla_i f(\hat{\beta}(\lambda_k))| \leq \alpha(\lambda_k - \lambda_{k+1}),$$

*for all $i \in \mathcal{G}_g$ for $g \in \mathcal{A}_g(\lambda_{k+1})$, then the candidate variable set,*

$$\mathcal{C}_v(\lambda_{k+1}) = \{i \in \mathcal{G}_g \text{ for } g \in \mathcal{A}_g(\lambda_{k+1}) :$$
$$|\nabla_i f(\hat{\beta}(\lambda_k))| > \alpha(2\lambda_{k+1} - \lambda_k)\},$$

*is such that $\mathcal{A}_v(\lambda_{k+1}) \subset \mathcal{C}_v(\lambda_{k+1})$.*

To derive Proposition 3.3, knowledge of $\mathcal{A}_g(\lambda_{k+1})$ is required, but this is unknown. By Proposition 3.2, $\mathcal{A}_g(\lambda_{k+1}) \subset \mathcal{C}_g(\lambda_{k+1})$, and so the candidate set is used in practice for applying Proposition 3.3. That is, the variable screening rule is applied to the candidate group set. Any violations caused by this replacement are checked for by the KKT checks, which are performed in any case (for any strong rule).

### 3.1.3. KKT CHECKS

The screening rules of DFR use several Lipschitz assumptions (Propositions 3.2 and 3.3), as well as replacing the active group set by the candidate group set for the variable screening step (Section 3.1.2). When these assumptions fail, the screening rules can incorrectly discard active variables. To protect against this, the KKT conditions are checked for each variable after screening. A KKT violation occurs for variable $i \in \mathcal{G}_g$ if

$$|S(\nabla_i f(\hat{\beta}(\lambda_{k+1})), \lambda_{k+1}(1-\alpha)\sqrt{p_g})| > \lambda_{k+1}\alpha, \quad (13)$$

where $S(a, b) = \text{sign}(a)(|a| - b)_+$ is the soft-thresholding operator (see Appendix A.2 for the derivation). A violating variable is added back into the optimization procedure (see Section 3.2).

### 3.1.4. PATH START

When fitting SGL along a path of values, $\lambda_1 \geq \ldots \geq \lambda_l \geq 0$, $\lambda_1$ is often chosen to be the exact point at which the first predictor becomes non-zero. By Ndiaye et al. (2016b) and using the dual norm from Equation 5, this value is given by

$$\lambda_1 = \|\nabla f(0)\|_{\text{sgl}}^* = \max_{g=1,\ldots,m} \tau_g^{-1} \|\nabla_g f(0)\|_{\epsilon_g}.$$

### 3.2. Algorithm

The DFR algorithm is based on the sparse-group strong screening framework, proposed by Feser & Evangelou (2025), and is shown in Algorithm A1. The algorithm has the following key steps for $\lambda_{k+1}$:

1. *Group screening*: find $\mathcal{C}_g(\lambda_{k+1})$ using Proposition 3.2.

2. *Variable screening*: find $\mathcal{C}_v(\lambda_{k+1})$ using Proposition 3.3 for $i \in \mathcal{G}_g \setminus \mathcal{A}_v(\lambda_k), g \in \mathcal{C}_g(\lambda_{k+1})$.

3. *Optimization*: Compute $\hat{\beta}_{\mathcal{O}_v}(\lambda_{k+1})$ using the *optimization set* $\mathcal{O}_v = \mathcal{C}_v(\lambda_{k+1}) \cup \mathcal{A}_v(\lambda_k)$. Perform KKT checks to identify any violations (Section 3.1.3) and add offending variables into $\mathcal{O}_v$. Repeat this step until no violations.

The two main computational costs of the algorithm are the calculation of the solution, $\hat{\beta}$, and the evaluation of the

$\epsilon$-norm. The former depends on the fitting algorithm, as this framework is applicable for any SGL fitting algorithm, with proximal and descent algorithms typically having complexities of $O(tp^2)$, for $t$ iterations (Zhao & Huo, 2023). The latter has a worst-case cost of $O(p_g \log p_g)$ (Ndiaye et al., 2016a). Tables A3 and A4 in the Appendix provide a computational breakdown of the components of DFR.

### 3.3. Adaptive Sparse-group Lasso

The *Adaptive Sparse-group Lasso* (aSGL) applies adaptive shrinkage in a sparse-group setting, achieving the oracle property in a double-asymptotic framework, and has the norm (Poignard, 2020; Mendez-Civieta et al., 2021)

$$\|\beta\|_{\text{asgl}} = \alpha \sum_{i=1}^{p} v_i |\beta_i| + (1-\alpha) \sum_{g=1}^{m} w_g \sqrt{p_g} \|\beta^{(g)}\|_2, \quad (14)$$

where $v_i$ and $w_g$ are adaptive weights (described in Appendix B.3). aSGL has a less straightforward connection to the $\epsilon$-norm that allows for the derivation of screening rules (Proposition 3.4).

**Proposition 3.4.** *The aSGL norm (Equation 14) can be expressed as the $\epsilon$-norm by*

$$\|\beta\|_{asgl} = \sum_{g=1}^{m} \gamma_g \|\beta^{(g)}\|_{\epsilon_g'}^*, \text{ where} \quad (15)$$

$$\gamma_g = \alpha \|v^{(g)}\|_1 - \frac{\alpha}{\|\hat{\beta}^{(g)}\|_1} \sum_{i,j \in \mathcal{G}_g, i \neq j} v_j |\hat{\beta}_i|$$
$$+ (1-\alpha) w_g \sqrt{p_g},$$
$$\epsilon_g' = \gamma_g^{-1}(1-\alpha) w_g \sqrt{p_g}.$$

*Proof of Proposition 3.4.* Splitting up the summation term in the variable norm yields

$$\alpha \sum_{i=1}^{p} v_i |\beta_i| = \alpha \sum_{g=1}^{m} \sum_{i \in \mathcal{G}_g} v_i |\beta_i|$$
$$= \alpha \sum_{g=1}^{m} \|\beta^{(g)}\|_1 \left( \|v^{(g)}\|_1 - \frac{\sum_{i,j \in \mathcal{G}_g, i \neq j} v_j |\beta_i|}{\|\beta^{(g)}\|_1} \right).$$

This allows the aSGL norm to be written in terms of the groups

$$\|\beta\|_{\text{asgl}} = \sum_{g=1}^{m} \left[ \left( \|v^{(g)}\|_1 - \frac{\sum_{i,j \in \mathcal{G}_g, i \neq j} v_j |\beta_i|}{\|\beta^{(g)}\|_1} \right) \alpha \|\beta^{(g)}\|_1 \right.$$
$$\left. + (1-\alpha) w_g \sqrt{p_g} \|\beta^{(g)}\|_2 \right]. \quad (16)$$

Setting

$$\gamma_g = \alpha \|v^{(g)}\|_1 - \frac{\alpha \sum_{i,j \in \mathcal{G}_g, i \neq j} v_j |\beta_i|}{\|\beta^{(g)}\|_1} + (1-\alpha) w_g \sqrt{p_g},$$
$$\epsilon_g' = \frac{(1-\alpha) w_g \sqrt{p_g}}{\gamma_g},$$

allows Equation 16 to be written in terms of the $\epsilon$-norm

$$\|\beta\|_{\text{asgl}} = \sum_{g=1}^{m} \gamma_g \|\beta^{(g)}\|_{\epsilon_g'}^*.$$

$\square$

Through this connection, the aSGL norm admits a direct link to SGL (Equation 3), allowing the DFR-SGL rules to be used for aSGL by replacing $\tau_g$ with $\gamma_g$ and $\epsilon_g$ with $\epsilon_g'$ in the $\epsilon$-norm (Appendices B.2.1 and B.2.2). The connection is further sharpened theoretically: Lemma 3.5 guarantees that the $\gamma_g$ term in Equation 15 will always exist, even under inactive groups. Lemma 3.6 guarantees that the representation of aSGL as the $\epsilon$-norm correctly reduces to the SGL $\epsilon$-norm representation under constant weights.

**Lemma 3.5.** *Under an inactive group $g \notin \mathcal{A}_g$, i.e. $\beta^{(g)} \equiv 0$, the $\gamma_g$ term in Equation 15 exists.*

**Lemma 3.6.** *Under $v \equiv 1$ and $w \equiv 1$ in Equation 15, for each $g \in [m]$, $\gamma_g = \tau_g$ and $\epsilon_g' = \epsilon_g$.*

Algorithm A1 is also applicable for aSGL, using the corresponding aSGL equations (Algorithm A2).

#### 3.3.1. KKT CHECKS

The KKT checks for aSGL are also similar to those for SGL (Section 3.1.3): a KKT violation occurs for a variable $i \in \mathcal{G}_g$ if

$$|S(\nabla_i f(\hat{\beta}(\lambda_{k+1})), \lambda_{k+1}(1-\alpha) w_g \sqrt{p_g})| > \lambda_{k+1} v_i \alpha. \quad (17)$$

#### 3.3.2. PATH START

To find the path start for aSGL, the dual norm cannot be used, since all groups are zero at this point. As a result, $\gamma_g$ exists only in limit for $\beta^{(g)} \equiv 0$ (Lemma 3.5). A solution can instead be found using a similar approach to that of Simon et al. (2013) for SGL, where the point is found by solving the piecewise quadratic, for each $g \in [m]$,

$$\left\| S\left( X^{(g)\top} y/n, \lambda_g v^{(g)} \alpha \right) \right\|_2^2 - p_g w_g^2 (1-\alpha)^2 \lambda_g^2 = 0,$$

where $X^{(g)} \in \mathbb{R}^{n \times p_g}$ is the design matrix for only group $g$ and $v^{(g)} \in \mathbb{R}^{p_g}$ contains the penalty weights for the variables in group $g$. Then, choosing $\lambda_1 = \max_g \lambda_g$ gives the path start point.

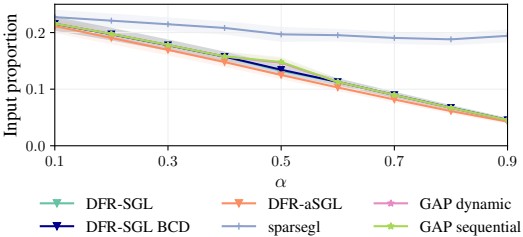

Figure 2: The input proportion for the screening methods applied to synthetic data, as a function of $\alpha$, with $95\%$ confidence intervals.

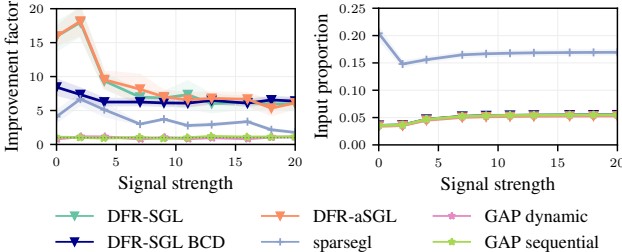

Figure 3: The improvement factor (left) and input proportion (right) for the screening methods applied to synthetic data, as a function of the signal strength, with $95\%$ confidence intervals.

# 4. Numerical Results

In this section, the efficiency and robustness of DFR is evaluated through the analysis of synthetic data that capture different data characteristics. As the purpose of screening rules is to reduce the dimensionality of the input space and, as a result, reduce the computational cost, the following two metrics are used for evaluation:

- *Improvement factor = no screen time / screen time*, quantifies by how many orders the screening has improved the fitting time (high is best).

- *Input proportion = $\mathcal{O}_v/p$*, measures how much of the input space was used in the optimization (low is best).

DFR is compared with the existing SGL screening rules sparsegl (Liang et al., 2022) and GAP safe (Ndiaye et al., 2016a). sparsegl, in contrast to DFR, performs only a single layer of group screening. This rule is also based on the strong framework of Tibshirani et al. (2010), but uses a different Lipschitz assumption, which applies only the $\ell_2$ group norm, rather than the full SGL norm (as is the case for DFR). On the other hand, GAP safe is an exact feature reduction method for SGL that can be implemented dynamically or sequentially under linear regression (Ndiaye et al., 2016a). GAP safe has many different implementation forms, and we are presenting the two versions that provided the best results in our studies. Appendix C provides detailed descriptions of these two methods with Table A1 showing a summary of all rules considered.

Throughout the analyses, the SGL optimization for DFR and sparsegl is performed using the Adaptive Three Operator Splitting (ATOS) (Pedregosa & Gidel, 2018) algorithm, as it can be easily adapted for use with different sparse-group penalties (by simply swapping out the proximal operators), providing flexibility. However, DFR can be used with any fitting algorithm, including Block Coordinate Descent (BCD) (Qin et al., 2013).

## 4.1. Analysis of Synthetic Data

The synthetic data was generated using a linear model, $y = \mathbf{X}\beta + \epsilon$, where $\mathbf{X} \sim \mathcal{N}(\mathbf{0}, \mathbf{\Sigma}) \in \mathbb{R}^{200 \times 1000}$, with noise $\epsilon \sim \mathcal{N}(0, 1)$ and where $\beta$ is a sparse vector with the signal sampled from $\mathcal{N}(0, 4)$ (signal strength of zero). For $\mathbf{X}$, correlation was applied inside each group, such that $\Sigma_{i,j} = \rho = 0.3$ for $i$ and $j$ in the same group. The variables were placed in even groups of sizes 20, with $0.2$ group sparsity proportion ($20\%$ of the groups are active) and $0.2$ variable sparsity proportion inside the active groups. The models were fit along a 50-length path, starting at $\lambda_1 = \lambda_{\max}$ (as defined by Section 3.1.4 for SGL and Section 3.3.2 for aSGL), and terminating at $0.1\lambda_1$. Each simulation case was repeated 100 times and the results are averaged across these repetitions, unless otherwise stated. Simulation and model implementation information can be found in Table D.2.

**Comparison to GAP Safe Rules**  Comparing DFR to the GAP safe rules, under both varying $\alpha$ and signal strength, it is evident that the improvement factor is significantly superior for DFR compared to both the dynamic and sequential GAP rules (Figures 1 and 3). In fact, although the input proportion of DFR and GAP safe are of similar levels (Figures 2 and 3), the cost of calculating safe regions appears to nullify any gain in dimensionality reduction. This comparison shows that the two reduction approaches (heuristic vs exact) arrive at very similar results (the screened sets), but DFR achieves this with greater computational efficiency.

For all values of $\alpha$, DFR considerably reduces the input space, with the screening efficiency a linearly decreasing function (Figure 2). Under values of $\alpha$ close to zero, SGL is forced to pick more variables within a group as active, limiting the potential reduction of the input space. In such scenarios, the second screening layer becomes less crucial, evidenced by the similar performances of all approaches. Approaching the commonly used value of $\alpha = 0.95$ shows the clear strengths of DFR. The screening methods are all relatively unaffected by the signal strength (Figure 3).

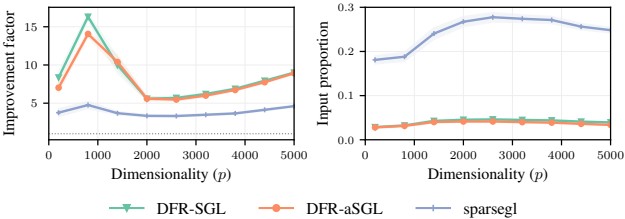

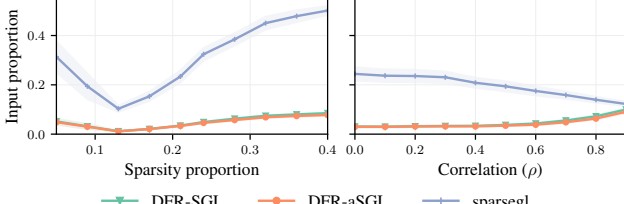

Figure 4: The improvement factor (left) and input proportion (right) for the strong rules applied to synthetic data, as a function of $p$, with $95\%$ confidence intervals.

Figure 6: The input proportion for the strong rules applied to synthetic data, as a function of the sparsity proportion (left) and data correlation (right), with $95\%$ confidence intervals.

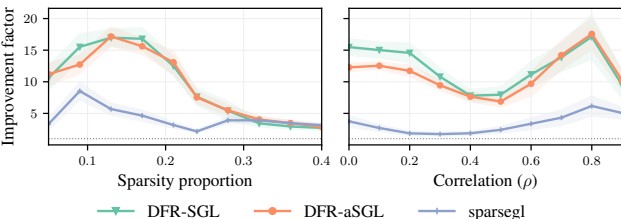

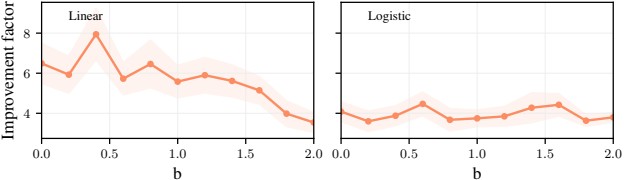

Figure 5: The improvement factor for the strong rules applied to synthetic data, as a function of the sparsity proportion (left) and data correlation (right), with $95\%$ confidence intervals.

Figure 7: The improvement factor of DFR-aSGL under different weights $b_1 = b_2$, shown for the linear (left) and logistic (right) models, with $95\%$ confidence intervals.

$p$, few large groups dominate, limiting the potential to screen out groups, while for large $p$, many small groups reduce the impact of group screening.

Additional explorations of increased dimensionality included the analysis of interaction terms (Table 1 and Appendix D.5). In this setting, all possible interactions of order 2 and 3 within each group were included. DFR provides large computational savings when fitting interactions, especially compared to sparsegl, which under order 3 interactions provides only marginal improvements. These savings make it more feasible for sparse-group models to be used in interaction detection problems. Such challenges are frequently seen in the field of genetics, where gene-gene and gene-environment relationships are useful discoveries (D'Angelo et al., 2009; Zemlianskaia et al., 2022).

**Comparison to BCD** The GAP safe rules are implemented using BCD while DFR uses ATOS. In Figures 1 and 3, DFR has also been implemented using BCD, demonstrating similar performance to ATOS, showing that the choice of fitting algorithm has little impact on the gain from screening.

**Increasing Dimensionality** The benefits of DFR over sparsegl are observed under varying $p$ (Figure 4), peaking around $p = 1000$. With groups of fixed sizes of 20, their relative size proportion to $p$ changes, suggesting an optimal grouping regime around $p = 1000$ for screening. For small

**Robustness** For the remainder of the section, the variables were placed in $m = 22$ uneven groups of sizes in $[3, 100]$ to gain additional insights into the robustness of DFR. Earlier, DFR was found to be robust under varying signal strength and $\alpha$. We further observe that DFR is also robust to the data-generating parameters of signal sparsity (variable and group sparsity proportions varied together) and group correlation in $\mathbf{X}$ (Figures 5 and 6).

A clear benefit of DFR over sparsegl is observed under sparse signals. Screening rules generally have a greater impact as the signal becomes sparser. However, once the signal saturates, their effectiveness declines, leading to similar performance across approaches. Under varying correlation, DFR is more successful at reducing the input space when

Table 1: The improvement factor for the strong rules applied to synthetic interaction data under the linear model, with standard errors. The parameters of the data were set as $p = 400, n = 80$, and $m = 52$ groups of sizes in $[3, 15]$. The interaction input dimensionality was $p_{O_2} = 2111$ and $p_{O_3} = 7338$, with no interaction hierarchy imposed. The sparsity proportion of interaction variables was set to 0.3 (with the same signal as the marginal effects).

| Method | Interaction | |
| --- | --- | --- |
| | Order 2 | Order 3 |
| DFR-aSGL | $137.3 \pm 12.0$ | $54.0 \pm 10.7$ |
| DFR-SGL | $44.3 \pm 2.4$ | $23.6 \pm 3.1$ |
| sparsegl | $7.4 \pm 0.9$ | $1.2 \pm 0.3$ |

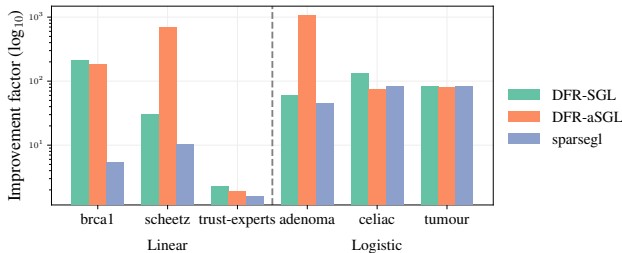

Figure 8: The improvement factor ($\log_{10}$ scale) of the strong rules applied to the six real datasets, split by model type.

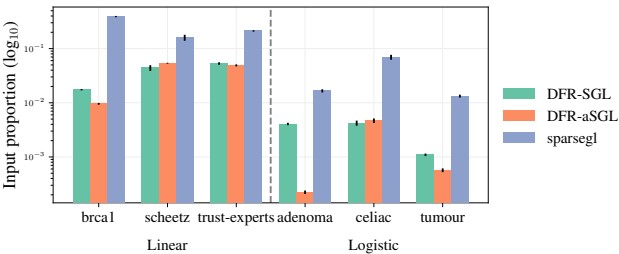

Figure 9: The input proportion ($\log_{10}$ scale) of the strong rules applied to the six real datasets, split by model type.

compared to sparsegl, especially under minor correlation. Under higher correlation, the models become less sparse, again resulting in reduced screening importance.

Similar to the case with even groups, for uneven groups, DFR remains relatively unaffected by the signal strength and the choice of $\alpha$, consistently achieving effective reduction. (Figures A2 and A3).

**Hyperparameter Tuning and Cross-Validation**   The performance of DFR-aSGL was found to be robust under different values of hyperparameters $b_1$ and $b_2$ (Figure 7), which are used to define the adaptive weights (Appendix B.3).

The efficiency and robustness of DFR across different hyperparameters $(\alpha, b_1, b_2)$ make it a promising tool for enabling approaches like cross-validation (CV) to tune all SGL and aSGL hyperparameters, which is rarely done in practice. Applying DFR with CV yields substantial computational savings (Table A11). These findings highlight DFR's value in facilitating expanded tuning regimes for SGL and aSGL.

**Logistic Model**   DFR is also effective and robust for logistic models (Figures A7, A8, A9, and A10; see Appendix D.6 for further results and description of the data generation).

**KKT Violations**   KKT violations for DFR are very rare. Across all experiments with linear models, DFR-SGL had only a single KKT violation (Table A9). Violations were more common for DFR-aSGL and sparsegl, but still infre-

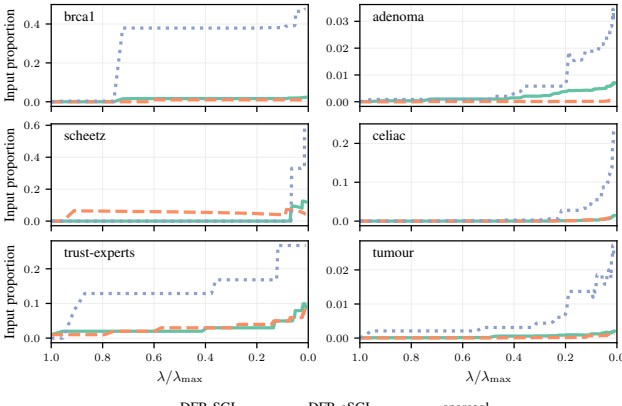

Figure 10: The input proportion as a function of the shrinkage path for the strong rules applied to the real datasets.

quent. Note that DFR-aSGL violations refer to variable ones, and sparsegl to group ones, making it more likely to have a variable violation. The elevated number of KKT violations for sparsegl suggests that the group Lipschitz assumption of DFR-SGL is more robust.

One possible explanation for the increased KKT violations of DFR-aSGL lies in the role of the Lipschitz assumptions. Unlike SGL, the adaptive penalties in aSGL introduce additional dependencies on hyperparameters into the Lipschitz assumptions. This extra dependence on hyperparameters has the potential to lead to violations.

## 5. Real Data Analysis

The efficiency of DFR is further evaluated through the analysis of six real datasets with different characteristics, including response type and dimensionality. Three of the datasets, *brca1*, *scheetz*, and *trust-experts*, have continuous responses, so are fit using an SGL linear model. The former two were also analyzed with regards to screening rules in Larsson & Wallin (2024), and the latter in Liang et al. (2022).

The other three datasets, *adenoma*, *celiac*, and *tumour*, have binary responses, so an SGL logistic model is used. The *trust-experts* dataset is low-dimensional, and the other five are high-dimensional. The models were fit along a 100-length path, terminating at $0.2\lambda_1$, where $\lambda_1$ generates the null model. More information on the datasets and model implementation is provided in Table D.2 and Appendix E.

For all datasets, DFR outperforms sparsegl at reducing the computational cost (Figure 8) and input dimension (Figure 9), as well as keeping the input proportion low along the whole path (Figure 10). Despite being most useful for high-dimensional data, even in the case of low-dimensional data (*trust-experts*), DFR improves fitting time.

DFR-aSGL performs very well for *scheetz* and *adenoma*, improving the computational cost by over 600 times. For the *scheetz* dataset, the aSGL model had more difficulty converging without screening compared to SGL, so DFR-aSGL offered a greater advantage over DFR-SGL. For *adenoma*, the active set for aSGL was smaller (Table A18), due to the increased penalization that comes with the adaptivity. However, despite the advantage of a smaller active set, we do still observe that DFR-aSGL was more efficient at reducing the optimization set, *with respect to the active set*.

DFR is observed to aid in mitigating convergence issues for both SGL and aSGL (Table A19). Across all datasets, DFR encountered no failed convergences. In contrast, sparsegl did not converge at several path points for both *adenoma* and *scheetz*. As sparsegl only screens groups, when a group enters the optimization set, sparsegl is forced to fit with the full group, which can contain noise variables. Applying no screening led to SGL not converging for *adenoma*, *scheetz*, and *tumour*. By drastically reducing the input space, convergence issues caused by large datasets are resolved, improving both computational cost and solution optimality.

## 6. Discussion

A novel feature reduction method for the sparse-group lasso and adaptive sparse-group lasso, called *Dual Feature Reduction* (DFR), has been introduced, derived using the dual norms of SGL and aSGL. DFR introduces the first bi-level strong screening rules for SGL and the first screening rules for aSGL. By applying two layers of reduction, DFR effectively reduces input dimensionality for optimisation and is computationally simpler than the GAP safe rules, which require iterative screening and fitting. In contrast, DFR screens only once per path point, so that it adds minimal computational overhead.

DFR first applies group-level screening, discarding inactive groups, followed by variable-level reduction, where inactive variables in active groups are removed. By discarding variables that are inactive at the optimal solution, DFR achieves significant computational savings, enabling the SGL family of models to scale more efficiently with increasing dimensionality and handle larger, more complex datasets. This gain comes at no cost, as the optimal solution is still achieved (Appendices D.4, D.6, and E.2). In fact, by reducing the input, instances were observed where DFR helped SGL and aSGL overcome convergence issues.

DFR proved robust across different data and model parameters, achieving drastic feature reduction under all scenarios considered. This consistently translated into large computational savings across both synthetic and real data. DFR outperformed all other screening approaches, establishing it as the state-of-the-art screening method for SGL and high-lighting the benefit of bi-level screening.

**Limitations** Several assumptions are required to perform two layers of feature reduction for DFR. Propositions 3.2 and 3.3 use Lipschitz assumptions which are consistent with the strong framework (Tibshirani et al., 2010). Any breach of assumptions is guarded against by KKT checks. Only a single KKT violation occurred for SGL across all our simulations and only very infrequently for aSGL. These assumptions are a limitation of any strong rule, although DFR carries additional assumptions over other strong rules, which are necessary for the second layer of screening.

**Code** DFR is implemented in the `dfr` R package (Feser, 2024), available on CRAN.

## Impact Statement

This paper aims to advance Machine Learning while ensuring no disadvantage to anyone. The proposed screening rules do not alter the solution but improve the accessibility of SGL and aSGL for researchers with limited computational resources.

## Acknowledgements

We would like to thank the anonymous reviewers for their valuable comments. This work was supported by the Engineering and Physical Sciences Research Council (EPSRC) through the Modern Statistics and Statistical Machine Learning (StatML) CDT programme, grant no. EP/S023151/1.

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

# Dual Feature Reduction for the Sparse-group Lasso and its Adaptive Variant: Supplementary Materials

## A. Sparse-group Lasso

### A.1. Theory

#### A.1.1. GROUP REDUCTION

*Proof of Proposition 3.1.* To prove the two sets are equivalent, we need to prove that for any $g \in [m]$ and $k \in [l-1]$, $g \in \mathcal{A}_g(\lambda_{k+1}) \iff g \in \mathcal{C}_g(\lambda_{k+1})$. We instead prove the contrapositive: $g \notin \mathcal{C}_g(\lambda_{k+1}) \iff g \notin \mathcal{A}_g(\lambda_{k+1})$. So,

$$\begin{aligned}
g \notin \mathcal{C}_g(\lambda_{k+1}) &\iff \|\nabla_g f(\hat{\beta}(\lambda_{k+1}))\|_{\epsilon_g} \leq \tau_g \lambda_{k+1}, &&\text{by definition of the candidate set} \\
&\iff -\nabla_g f(\hat{\beta}(\lambda_{k+1})) \in \tau_g \lambda_{k+1} \Theta^0_{g,k+1}, &&\text{as } \Theta^0_{g,k+1} = \left\{ x \in \mathbb{R}^{p_g} : \|x\|_{\epsilon_g} \leq 1 \right\} \\
&\iff \mathbf{0} \in \nabla_g f(\hat{\beta}(\lambda_{k+1})) + \tau_g \lambda_{k+1} \Theta^0_{g,k+1} \\
&\iff g \notin \mathcal{A}_g(\lambda_{k+1}), &&\text{by the KKT conditions (Equation 6).}
\end{aligned}$$

$\square$

*Proof of Proposition 3.2.* To prove the candidate set is a superset of the active set, we need to prove that for any $g \in [m]$ and $k \in [l-1]$, $g \in \mathcal{A}_g(\lambda_{k+1}) \implies g \in \mathcal{C}_g(\lambda_{k+1})$. We instead prove the contrapositive: $g \notin \mathcal{C}_g(\lambda_{k+1}) \implies g \notin \mathcal{A}_g(\lambda_{k+1})$. First, we rewrite the Lipschitz assumption as (using the reverse triangle inequality)

$$\begin{aligned}
\|\nabla_g f(\hat{\beta}(\lambda_{k+1}))\|_{\epsilon_g} - \|\nabla_g f(\hat{\beta}(\lambda_k))\|_{\epsilon_g} &\leq \|\nabla_g f(\hat{\beta}(\lambda_{k+1})) - \nabla_g f(\hat{\beta}(\lambda_k))\|_{\epsilon_g} \leq \tau_g |\lambda_{k+1} - \lambda_k| \\
&\implies \|\nabla_g f(\hat{\beta}(\lambda_{k+1}))\|_{\epsilon_g} \leq \|\nabla_g f(\hat{\beta}(\lambda_k))\|_{\epsilon_g} + \tau_g |\lambda_{k+1} - \lambda_k|.
\end{aligned} \tag{18}$$

Now, as $g \notin \mathcal{C}_g(\lambda_{k+1})$,

$$\|\nabla_g f(\hat{\beta}(\lambda_k))\|_{\epsilon_g} \leq \tau_g (2\lambda_{k+1} - \lambda_k).$$

Plugging this into Equation 18 yields

$$\begin{aligned}
&\|\nabla_g f(\hat{\beta}(\lambda_{k+1}))\|_{\epsilon_g} \leq \tau_g (2\lambda_{k+1} - \lambda_k) + \tau_g |\lambda_{k+1} - \lambda_k| \\
\implies &\|\nabla_g f(\hat{\beta}(\lambda_{k+1}))\|_{\epsilon_g} \leq \tau_g \lambda_{k+1} \\
\implies &-\nabla_g f(\hat{\beta}(\lambda_{k+1})) \in \tau_g \lambda_{k+1} \Theta^0_{g,k+1}, &&\text{as } \Theta^0_{g,k+1} = \left\{ x \in \mathbb{R}^{p_g} : \|x\|_{\epsilon_g} \leq 1 \right\} \\
\implies &\mathbf{0} \in \nabla_g f(\hat{\beta}(\lambda_{k+1})) + \tau_g \lambda_{k+1} \Theta^0_{g,k+1} \\
\implies &g \notin \mathcal{A}_g(\lambda_{k+1}), &&\text{by the KKT conditions (Equation 6).}
\end{aligned}$$

$\square$

#### A.1.2. VARIABLE REDUCTION

**Proposition A.1** (Theoretical SGL variable screening). *For SGL applied with any $\lambda_{k+1}, k \in [l-1]$, the candidate variable set,*

$$\mathcal{C}_v(\lambda_{k+1}) = \{i \in \mathcal{G}_g \text{ for } g \in \mathcal{A}_g(\lambda_{k+1}) : |\nabla_i f(\hat{\beta}(\lambda_{k+1}))| > \lambda_{k+1}\alpha\},$$

*is such that $\mathcal{C}_v(\lambda_{k+1}) = \mathcal{A}_v(\lambda_{k+1})$.*

*Proof of Proposition A.1.* The proof strategy is similar to that of Proposition 3.1. To prove the two sets are equivalent, we need to prove that for any $i \in \mathcal{G}_g$ such that $g \in \mathcal{A}_g$, and $k \in [l-1]$, $i \in \mathcal{A}_v(\lambda_{k+1}) \iff i \in \mathcal{C}_v(\lambda_{k+1})$. We instead prove the contrapositive: $i \notin \mathcal{C}_v(\lambda_{k+1}) \iff i \notin \mathcal{A}_v(\lambda_{k+1})$. So,

$$
\begin{aligned}
i \notin \mathcal{C}_v(\lambda_{k+1}) &\iff |\nabla_i f(\hat{\beta}(\lambda_{k+1}))| \le \lambda_{k+1}\alpha, &&\text{by definition of the candidate set} \\
&\iff -\nabla_v f(\hat{\beta}(\lambda_{k+1})) \in \lambda_{k+1}\alpha\Phi^0_{i,k+1}, &&\text{as } \Phi^0_{i,k+1} = \{x \in \mathbb{R} : |x| \le 1\}, \\
&&&\text{for } i \in \mathcal{G}_g, g \in \mathcal{A}_g(\lambda_{k+1}) \\
&\iff \mathbf{0} \in \nabla_v f(\hat{\beta}(\lambda_{k+1})) + \lambda_{k+1}\alpha\Phi^0_{i,k+1} \\
&\iff i \notin \mathcal{A}_v(\lambda_{k+1}), &&\text{by the KKT conditions (Equation 11).}
\end{aligned}
$$

$\square$

*Proof of Proposition 3.3.* The proof strategy is similar to that of Proposition 3.2. To prove the candidate set is a superset of the active set, we need to prove that for any $i \in \mathcal{G}_g$ such that $g \in \mathcal{A}_g$, and $k \in [l-1]$, $i \in \mathcal{A}_v(\lambda_{k+1}) \implies i \in \mathcal{C}_v(\lambda_{k+1})$. We instead prove the contrapositive: $i \notin \mathcal{C}_v(\lambda_{k+1}) \implies i \notin \mathcal{A}_v(\lambda_{k+1})$. First, we rewrite the Lipschitz assumption as (using the reverse triangle inequality)

$$|\nabla_i f(\hat{\beta}(\lambda_{k+1}))| \le |\nabla_i f(\hat{\beta}(\lambda_k))| + \alpha|\lambda_{k+1} - \lambda_k|. \tag{19}$$

Now, as $i \notin \mathcal{C}_v(\lambda_{k+1})$,

$$|\nabla_i f(\hat{\beta}(\lambda_k))| \le \alpha(2\lambda_{k+1} - \lambda_k).$$

Plugging this into Equation 19 yields

$$
\begin{aligned}
&|\nabla_i f(\hat{\beta}(\lambda_{k+1}))| \le \alpha\lambda_{k+1} \\
&\implies -\nabla_i f(\hat{\beta}(\lambda_{k+1})) \in \alpha\lambda_{k+1}\Phi^0_{i,k+1}, \ \text{as } \Phi^0_{i,k+1} = \{x \in \mathbb{R} : |x| \le 1\} \\
&\implies \mathbf{0} \in \nabla_i f(\hat{\beta}(\lambda_{k+1})) + \alpha\lambda_{k+1}\Phi^0_{i,k+1} \\
&\implies i \notin \mathcal{A}_v(\lambda_{k+1}), \qquad\qquad \text{by the KKT conditions (Equation 6).}
\end{aligned}
$$

$\square$

## A.2. KKT Checks

To determine whether a variable $i \in \mathcal{G}_g$ has been correctly discarded, the KKT stationarity conditions are checked. Equation 11 describes the condition under which a variable $i \in \mathcal{G}_g$ is inactive. Without specifying whether the group $g$ is inactive, this can be rewritten as (by the definition of $\Phi^0_{i,k+1}$)

$$|\nabla_i f(\hat{\beta}(\lambda_{k+1})) + \lambda_{k+1}(1-\alpha)\Psi^{(g)}_{i,k+1}| \le \lambda_{k+1}\alpha, \tag{20}$$

where $\Psi^{(g)}_{k+1} = \{x \in \mathbb{R}^{\sqrt{p_g}} : \|x\|_2 \le 1\}$ is the subgradient of the $\ell_2$ norm. To satisfy Equation 20, the unknown subdifferential, $\Psi^{(g)}_{i,k+1}$, is taken to be the minimum possible value. For $x \in \Psi^{(g)}_{k+1}$, we have that

$$
\begin{aligned}
\|x\|_2 \le 1 &\implies \sqrt{p_g}\|x\|_2 \le \sqrt{p_g} \\
&\implies \|x\|_1 \le \sqrt{p_g} \ \text{ by the inequality } \|x\|_1 \le \sqrt{p_g}\|x\|_2 \\
&\implies |x_i| \le \sqrt{p_g}.
\end{aligned}
$$

Hence, the values in the subdifferential are bounded by $\sqrt{p_g}$. We consider the following scenarios for Equation 20:

1. $\nabla_i f(\hat{\beta}(\lambda_{k+1})) > \lambda_{k+1}(1-\alpha)\sqrt{p_g}$: choose $x_i = -\sqrt{p_g}$.

2. $\nabla_i f(\hat{\beta}(\lambda_{k+1})) < -\lambda_{k+1}(1-\alpha)\sqrt{p_g}$: choose $x_i = \sqrt{p_g}$.

3. $\nabla_i f(\hat{\beta}(\lambda_{k+1})) \in [-\lambda_{k+1}(1-\alpha)\sqrt{p_g}, \lambda_{k+1}(1-\alpha)\sqrt{p_g}]$: choose $y_i = \frac{\nabla_i f(\hat{\beta}(\lambda_{k+1}))}{\lambda_{k+1}(1-\alpha)\sqrt{p_g}}$.

This allows Equation 20 to be expressed using the soft-thresholding operator as

$$|S(\nabla_i f(\hat{\beta}(\lambda_{k+1})), \lambda_{k+1}(1-\alpha)\sqrt{p_g})| \leq \lambda_{k+1}\alpha.$$

A similar derivation can be found in Simon et al. (2013) to derive conditions to check whether a group is active for SGL.

### A.3. Algorithm

---

**Algorithm A1** Dual Feature Reduction (DFR) for SGL

---

**Input:** $(\lambda_1, \ldots, \lambda_l) \in \mathbb{R}^l$, $\mathbf{X} \in \mathbb{R}^{n \times p}$, $y \in \mathbb{R}^n$, $\alpha \in [0, 1]$
compute $\hat{\beta}(\lambda_1)$ using Equation 1
**for** $k = 1$ **to** $l - 1$ **do**
    $\mathcal{C}_g(\lambda_{k+1}) \leftarrow$ candidate groups from Proposition 3.2
    $\mathcal{C}_v(\lambda_{k+1}) \leftarrow$ candidate variables from Proposition 3.3 for $i \in \mathcal{G}_g \setminus \mathcal{A}_v(\lambda_k), g \in \mathcal{C}_g(\lambda_{k+1})$
    $\mathcal{O}_v \leftarrow \mathcal{C}_v(\lambda_{k+1}) \cup \mathcal{A}_v(\lambda_k)$                    ▶ Optimization set
    compute $\hat{\beta}_i(\lambda_{k+1}), i \in \mathcal{O}_v$, using Equation 1
    $\mathcal{K}_v \leftarrow$ variable KKT violations for $i \notin \mathcal{O}_v$, using Equation 13         ▶ KKT check
    **while** card$(\mathcal{K}_v) > 0$ **do**
        $\mathcal{O}_v \leftarrow \mathcal{O}_v \cup \mathcal{K}_v$                    ▶ Optimization set
        compute $\hat{\beta}_i(\lambda_{k+1}), i \in \mathcal{O}_v$, using Equation 1
        $\mathcal{K}_v \leftarrow$ variable KKT violations for $i \notin \mathcal{O}_v$ using Equation 13         ▶ KKT check
    **end while**
**end for**
**Output:** $\hat{\beta}_{\text{sgl}}(\lambda_1), \ldots, \hat{\beta}_{\text{sgl}}(\lambda_l) \in \mathbb{R}^p$

---

### A.4. Reduction to (Adaptive) Lasso and (Adaptive) Group Lasso

Under $\alpha = 1$, SGL reduces to the lasso. In this case, no group screening occurs and the variable screening rule reduces to the lasso strong rule (Tibshirani et al., 2010):

$$|\nabla_i f(\hat{\beta}(\lambda_k))| \leq 2\lambda_{k+1} - \lambda_k.$$

Under $\alpha = 0$, SGL reduces to the group lasso. Under this scenario, the group screening reduces to the group lasso strong rule (Tibshirani et al., 2010):

$$\|\nabla_g f(\hat{\beta}(\lambda_k))\|_2 \leq \sqrt{p_g}(2\lambda_{k+1} - \lambda_k),$$

and no variable screening is performed. For aSGL, the rules reduce to the adaptive lasso and adaptive group lasso:

$$\text{Adaptive lasso: } |\nabla_i f(\hat{\beta}(\lambda_k))| \leq v_i(2\lambda_{k+1} - \lambda_k) \qquad \implies \hat{\beta}_i(\lambda_{k+1}) = 0.$$

$$\text{Adaptive group lasso: } \|\nabla_g f(\hat{\beta}(\lambda_k))\|_{\epsilon'_{g,1}} \leq w_g\sqrt{p_g}(2\lambda_{k+1} - \lambda_k) \implies \hat{\beta}^{(g)}(\lambda_{k+1}) \equiv \mathbf{0},$$

where $\epsilon'_{g,1}$ denotes the $\epsilon$-norm under $\epsilon'_g = 1$ (Equations 4 and 15).

# B. Adaptive Sparse-group Lasso

## B.1. Derivation of the Connection to $\epsilon$-norm

*Full proof of Proposition 3.4.* The adaptive SGL is given by

$$\|\beta\|_{\text{asgl}} = \alpha \sum_{i=1}^{p} v_i |\beta_i| + (1-\alpha) \sum_{g=1}^{m} w_g \sqrt{p_g} \|\beta^{(g)}\|_2.$$

The aim is to link this norm to the $\epsilon$-norm, in a similar way to SGL:

$$\|\beta\|_{\text{sgl}} = \sum_{g=1}^{m} (\alpha + (1-\alpha)\sqrt{p_g}) \|\beta^{(g)}\|_{\epsilon_g}^*.$$

Splitting up the summation term in the adaptive lasso norm yields

$$\alpha \sum_{i=1}^{p} v_i |\beta_i| = \alpha \sum_{g=1}^{m} \sum_{i \in \mathcal{G}_g} v_i |\beta_i|$$

$$= \alpha \sum_{g=1}^{m} \left( \sum_{j \in \mathcal{G}_g} v_j \sum_{i \in \mathcal{G}_g} |\beta_i| - \sum_{i,j \in \mathcal{G}_g, i \neq j} v_j |\beta_i| \right)$$

$$= \alpha \sum_{g=1}^{m} \left( \sum_{j \in \mathcal{G}_g} v_j \sum_{i \in \mathcal{G}_g} |\beta_i| - \frac{\sum_{i,j \in \mathcal{G}_g, i \neq j} v_j |\beta_i|}{\sum_{i \in \mathcal{G}_g} |\beta_i|} \sum_{i \in \mathcal{G}_g} |\beta_i| \right)$$

$$= \alpha \sum_{g=1}^{m} \sum_{i \in \mathcal{G}_g} |\beta_i| \left( \sum_{j \in \mathcal{G}_g} v_j - \frac{\sum_{i,j \in \mathcal{G}_g, i \neq j} v_j |\beta_i|}{\sum_{i \in \mathcal{G}_g} |\beta_i|} \right)$$

$$= \alpha \sum_{g=1}^{m} \|\beta^{(g)}\|_1 \left( \|v^{(g)}\|_1 - \frac{\sum_{i,j \in \mathcal{G}_g, i \neq j} v_j |\beta_i|}{\|\beta^{(g)}\|_1} \right).$$

Hence

$$\|\beta\|_{\text{asgl}} = \alpha \sum_{i=1}^{p} v_i |\beta_i| + (1-\alpha) \sum_{g=1}^{m} w_g \sqrt{p_g} \|\beta^{(g)}\|_2$$

$$= \sum_{g=1}^{m} \left[ \left( \|v^{(g)}\|_1 - \frac{\sum_{i,j \in \mathcal{G}_g, i \neq j} v_j |\beta_i|}{\|\beta^{(g)}\|_1} \right) \alpha \|\beta^{(g)}\|_1 + (1-\alpha) w_g \sqrt{p_g} \|\beta^{(g)}\|_2 \right]. \tag{21}$$

Setting

$$\gamma_g = \alpha \|v^{(g)}\|_1 - \frac{\alpha \sum_{i,j \in \mathcal{G}_g, i \neq j} v_j |\beta_i|}{\|\beta^{(g)}\|_1} + (1-\alpha) w_g \sqrt{p_g},$$

simplifies Equation 21 to

$$\|\beta\|_{\text{asgl}} = \sum_{g=1}^{m} \gamma_g \left[ \left( \frac{\gamma_g - (1-\alpha) w_g \sqrt{p_g}}{\gamma_g} \right) \|\beta^{(g)}\|_1 + \left( \frac{(1-\alpha) w_g \sqrt{p_g}}{\gamma_g} \right) \|\beta^{(g)}\|_2 \right]. \tag{22}$$

Setting

$$\epsilon_g' = \frac{(1-\alpha) w_g \sqrt{p_g}}{\gamma_g},$$

allows Equation 22 to be written in terms of the $\epsilon$-norm

$$\|\beta\|_{\text{asgl}} = \sum_{g=1}^{m} \gamma_g \left[ (1-\epsilon_g') \|\beta^{(g)}\|_1 + \epsilon_g' \|\beta^{(g)}\|_2 \right] = \sum_{g=1}^{m} \gamma_g \|\beta^{(g)}\|_{\epsilon_g'}^*.$$

$$\square$$

### B.1.1. LEMMA PROOFS FOR THE CONNECTION TO THE $\epsilon$-NORM

*Proof of Lemma 3.5.* Under $\beta^{(g)} \equiv \mathbf{0}$ for a group $g \notin \mathcal{A}_g$, the middle term in $\gamma_g$ becomes

$$\lim_{\beta^{(g)} \to \mathbf{0}} \left( \frac{\alpha \sum_{i,j \in \mathcal{G}_g, i \neq j} v_j |\beta_i|}{\|\beta^{(g)}\|_1} \right) = \frac{\alpha(p_g - 1)}{p_g} \sum_{i=1}^{p_g} v_i,$$

so that $\gamma_g$ still exists. This can be observed by using L'Hôpital's rule and noting that for $i \in \mathcal{G}_g$,

$$\frac{\partial}{\partial \beta_i} \sum_{i \neq j} v_j |\beta_i| = \sum_{i \neq j} v_j, \quad \frac{\partial}{\partial \beta_i} \|\beta^{(g)}\|_1 = 1.$$

$\square$

*Proof of Lemma 3.6.* Under $v \equiv \mathbf{1}$ and $w \equiv \mathbf{1}$, note that

$$\gamma_g = \alpha \left( p_g - \frac{\sum_{i,j \in \mathcal{G}_g, i \neq j} v_j |\beta_i|}{\|\beta^{(g)}\|_1} \right) + (1 - \alpha)\sqrt{p_g}$$

$$= \alpha \left( p_g - \frac{(p_g - 1)\|\beta^{(g)}\|_1}{\|\beta^{(g)}\|_1} \right) + (1 - \alpha)\sqrt{p_g}$$

$$= \alpha + (1 - \alpha)\sqrt{p_g} = \tau_g.$$

To understand the cross summation term, note that we are summing over each $\beta$ term $p_g - 1$ times, as the matching indices are removed, that is (for ease of notation, we consider $\mathcal{G}_1$ so that the indexing here is reset from 1)

$$\sum_{i,j \in \mathcal{G}_1, i \neq j} v_j |\beta_i| = |\beta_1| v_2 + \ldots + |\beta_1| v_{p_1} + |\beta_2| v_1 + \ldots + |\beta_2| v_{p_1} + \ldots + |\beta_{p_1}| v_{p_1 - 1}$$

$$= (p_1 - 1)|\beta_1| + \ldots + (p_1 - 1)|\beta_{p_1}|, \text{ by setting } v_j = 1, \forall j \in \mathcal{G}_1, \text{ for SGL}$$

$$= (p_1 - 1) \sum_{i \in \mathcal{G}_1} |\beta_i| = (p_1 - 1)\|\beta^{(1)}\|_1.$$

Hence, using $w_g = 1$ and $\tau_g = \alpha + (1 - \alpha)\sqrt{p_g}$,

$$\epsilon'_g = \frac{(1 - \alpha)w_g\sqrt{p_g}}{\gamma_g} = \frac{(1 - \alpha)\sqrt{p_g}}{\tau_g} = \frac{\tau_g - \alpha}{\tau_g} = \epsilon_g.$$

$\square$

## B.2. Theory

### B.2.1. GROUP SCREENING

To derive the group screening rule for aSGL, we compare the formulations of SGL and aSGL in terms of the $\epsilon$-norm (Equations 3 and 15):

$$\|\beta\|_{\text{sgl}} = \sum_{g=1}^{m} \tau_g \|\beta^{(g)}\|^*_{\epsilon_g}, \quad \|\beta\|_{\text{asgl}} = \sum_{g=1}^{m} \gamma_g \|\beta^{(g)}\|^*_{\epsilon'_g}.$$

Therefore, the derivation for the group screening rule for aSGL is identical to that of SGL (Section 3.1.1) replacing $\tau_g$ with $\gamma_g$ and $\|\cdot\|_{\epsilon_g}$ with $\|\cdot\|_{\epsilon'_g}$. The group screening rule is given by: discard a group $g$ if

$$\|\nabla_g f(\hat{\beta}(\lambda_k))\|_{\epsilon'_g} \leq \gamma_g(2\lambda_{k+1} - \lambda_k), \tag{23}$$

and is formalized in Propositions B.1 and B.2.

**Proposition B.1** (Theoretical aSGL group screening). *For aSGL applied with any $\lambda_{k+1}, k \in [l-1]$, the candidate group set,*

$$\mathcal{C}_g(\lambda_{k+1}) = \{g \in [m] : \|\nabla_g f(\hat{\beta}(\lambda_{k+1}))\|_{\epsilon'_g} > \gamma_g \lambda_{k+1}\},$$

*is such that $\mathcal{C}_g(\lambda_{k+1}) = \mathcal{A}_g(\lambda_{k+1})$.*

*Proof.* The proof is identical to that of Proposition 3.1 replacing $\tau_g$ with $\gamma_g$ and $\|\cdot\|_{\epsilon_g}$ with $\|\cdot\|_{\epsilon'_g}$ (see Appendix A.1.1). □

**Proposition B.2** (DFR-aSGL group screening). *For aSGL applied with any $\lambda_{k+1}, k \in [l-1]$, assuming that*

$$\|\nabla_g f(\hat{\beta}(\lambda_{k+1})) - \nabla_g f(\hat{\beta}(\lambda_k))\|_{\epsilon'_g} \leq \gamma_g |\lambda_{k+1} - \lambda_k|,$$

*for all $g \in [m]$, then the candidate group set,*

$$\mathcal{C}_g(\lambda_{k+1}) = \{g \in [m] : \|\nabla_g f(\hat{\beta}(\lambda_k))\|_{\epsilon'_g} > \gamma_g(2\lambda_{k+1} - \lambda_k)\},$$

*is such that $\mathcal{A}_g(\lambda_{k+1}) \subset \mathcal{C}_g(\lambda_{k+1})$.*

*Proof.* The proof is identical to that of Proposition 3.2 replacing $\tau_g$ with $\gamma_g$ and $\|\cdot\|_{\epsilon_g}$ with $\|\cdot\|_{\epsilon'_g}$ (see Appendix A.1.1). □

### B.2.2. VARIABLE SCREENING

The construction of the variable screening rule for aSGL is very similar to that of SGL (Section 3.1.2). The KKT stationary conditions for aSGL for an inactive variable in an active group are (in comparison to Equation 12 for SGL)

$$-\nabla_i f(\hat{\beta}(\lambda_{k+1})) \in \lambda_{k+1}\alpha v_i \Phi^0_{i,k+1}.$$

Therefore, the derivation of the rule is identical, replacing $\alpha$ with $\alpha v_i$. The variable screening rule is given by: discard a variable $i$ if

$$|\nabla_i f(\hat{\beta}(\lambda_k))| \leq \alpha v_i(2\lambda_{k+1} - \lambda_k), \tag{24}$$

and is formalized in Propositions B.3 and B.4.

**Proposition B.3** (Theoretical aSGL variable screening). *For aSGL applied with any $\lambda_{k+1}, k \in [l-1]$, the candidate variable set,*

$$\mathcal{C}_v(\lambda_{k+1}) = \{i \in \mathcal{G}_g \text{ for } g \in \mathcal{A}_g(\lambda_{k+1}) : |\nabla_i f(\hat{\beta}(\lambda_{k+1}))| > \lambda_{k+1}\alpha v_i\},$$

*is such that $\mathcal{C}_v(\lambda_{k+1}) = \mathcal{A}_v(\lambda_{k+1})$.*

*Proof.* The proof is identical to that of Proposition A.1 replacing $\alpha$ with $\alpha v_i$ (see Appendix A.1.2). □

**Proposition B.4** (DFR-aSGL variable screening). *For aSGL applied with any $\lambda_{k+1}, k \in [l-1]$, assuming that*

$$|\nabla_i f(\hat{\beta}(\lambda_{k+1})) - \nabla_i f(\hat{\beta}(\lambda_k))| \leq \alpha v_i(\lambda_k - \lambda_{k+1}),$$

*for all $i \in \mathcal{G}_g$ for $g \in \mathcal{A}_g(\lambda_{k+1})$, then the variable candidate set,*

$$\mathcal{C}_v(\lambda_{k+1}) = \{i \in \mathcal{G}_g \text{ for } g \in \mathcal{A}_g(\lambda_{k+1}) : |\nabla_i f(\hat{\beta}(\lambda_k))| > \alpha v_i(2\lambda_{k+1} - \lambda_k)\},$$

*is such that $\mathcal{A}_v(\lambda_{k+1}) \subset \mathcal{C}_v(\lambda_{k+1})$.*

*Proof.* The proof is identical to that of Proposition 3.3 replacing $\alpha$ with $\alpha v_i$ (see Appendix A.1.2). □

## B.3. Choice of Adaptive Weights

The adaptive weights are chosen according to Mendez-Civieta et al. (2021) as

$$v_i = \frac{1}{|q_{1i}|^{b_1}}, w_g = \frac{1}{\|q_1^{(g)}\|_2^{b_2}},$$

where $q_1$ is the first principal component from performing principal component analysis on $\mathbf{X}$ and $b_1, b_2$ are chosen by the user, often in the range $[0, 2]$. The weights are shown for $b_1 = b_2 = 0.1$ in Figure A1.

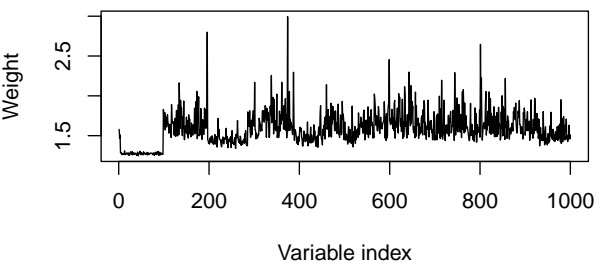
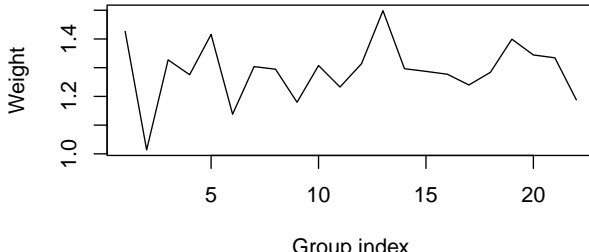

Figure A1: The weights, $(v, w)$, for aSGL, used in Figure 5 (right), where $p = 1000, n = 200, m = 22, \rho = 0.3, b_1 = b_2 = 0.1$, and $\alpha = 0.95$.

## B.4. Algorithm

---

**Algorithm A2** Dual Feature Reduction (DFR) for aSGL

---

**Input:** $(\lambda_1, \ldots, \lambda_l) \in \mathbb{R}^l, \mathbf{X} \in \mathbb{R}^{n \times p}, y \in \mathbb{R}^n, \alpha \in [0, 1]$

compute $\hat{\beta}(\lambda_1)$ using Equation 1, replacing the SGL norm with Equation 14

**for** $k = 1$ **to** $l - 1$ **do**

$\quad \mathcal{C}_g(\lambda_{k+1}) \leftarrow$ candidate groups from Equation 23

$\quad \mathcal{C}_v(\lambda_{k+1}) \leftarrow$ candidate variables from Equation 24 for $i \in \mathcal{G}_g \setminus \mathcal{A}_v(\lambda_k), g \in \mathcal{C}_g(\lambda_{k+1})$

$\quad \mathcal{O}_v \leftarrow \mathcal{C}_v(\lambda_{k+1}) \cup \mathcal{A}_v(\lambda_k)$ ▶ Optimization set

$\quad$ compute $\hat{\beta}_i(\lambda_{k+1}), i \in \mathcal{O}_v$, using Equation 1, replacing the SGL norm with Equation 14

$\quad \mathcal{K}_v \leftarrow$ variable KKT violations for $i \notin \mathcal{O}_v$, using Equation 17 ▶ KKT check

$\quad$ **while** $\text{card}(\mathcal{K}_v) > 0$ **do**

$\quad\quad \mathcal{O}_v \leftarrow \mathcal{O}_v \cup \mathcal{K}_v$ ▶ Optimization set

$\quad\quad$ compute $\hat{\beta}_i(\lambda_{k+1}), i \in \mathcal{O}_v$, using Equation 1, replacing the SGL norm with Equation 14

$\quad\quad \mathcal{K}_v \leftarrow$ variable KKT violations for $i \notin \mathcal{O}_v$, using Equation 17 ▶ KKT check

$\quad$ **end while**

**end for**

**Output:** $\hat{\beta}_{\text{asgl}}(\lambda_1), \ldots, \hat{\beta}_{\text{asgl}}(\lambda_l) \in \mathbb{R}^p$

---

## C. Competitive Feature Reduction Approaches

Table A1: A summary of the four screening rules for SGL considered.

| | | Rules (discard if true) | |
|---|---|---|---|
| Method | Type | Variable | Group |
| DFR-aSGL | Heuristic | $\|\nabla_i f(\hat{\beta}(\lambda_k))\| \leq \alpha v_i(2\lambda_{k+1} - \lambda_k)$ | $\|\nabla_g f(\hat{\beta}(\lambda_k))\|_{\epsilon'_g} \leq \gamma_g(2\lambda_{k+1} - \lambda_k)$ |
| DFR-SGL | Heuristic | $\|\nabla_i f(\hat{\beta}(\lambda_k))\| \leq \alpha(2\lambda_{k+1} - \lambda_k)$ | $\|\nabla_g f(\hat{\beta}(\lambda_k))\|_{\epsilon_g} \leq \tau_g(2\lambda_{k+1} - \lambda_k)$ |
| sparsegl | Heuristic | - | $\|S(\nabla_g f(\hat{\beta}(\lambda_k)), \lambda_k\alpha)\|_2 \leq \sqrt{p_g}(1-\alpha)(2\lambda_{k+1} - \lambda_k)$ |
| GAP safe | Exact | $\|X_i^\top \Theta_c\| + r\|X_i\|_2 < \tau$ | $\mathcal{T}_g < (1-\alpha)\sqrt{p_g}$ |

**sparsegl**  sparsegl is a screening rule proposed by Liang et al. (2022) and performs a single layer of group screening. The rule is based on the strong screening framework (Tibshirani et al., 2010) and the first order condition derived in Simon et al. (2013), i.e., that a group $g \in [m]$ is inactive if

$$\|S(\nabla_g f(\hat{\beta}(\lambda_{k+1})), \lambda_{k+1}\alpha)\|_2 \leq \sqrt{p_g}(1-\alpha)\lambda_{k+1}.$$

As the gradient at $k+1$ is not available, the following Lipschitz assumption on the $\ell_2$ norm is used:

$$\|S(\nabla_g f(\hat{\beta}(\lambda_{k+1})), \lambda_{k+1}\alpha) - S(\nabla_g f(\hat{\beta}(\lambda_k)), \lambda_k\alpha)\|_2 \leq \sqrt{p_g}(1-\alpha)|\lambda_{k+1} - \lambda_k|.$$

This leads to the sparsegl screening rule (via the triangle inequality): discard a group $g$ if

$$\|S(\nabla_g f(\hat{\beta}(\lambda_k)), \lambda_k\alpha)\|_2 \leq \sqrt{p_g}(1-\alpha)(2\lambda_{k+1} - \lambda_k).$$

This screening rule uses a different Lipschitz assumption at the group-level (DFR: Equation 9), which in turn leads to a different group-level rule (DFR: Equation 10). Our Lipschitz assumption is more consistent with the work of Tibshirani et al. (2010), as the assumption is with regards to the dual norm of the full SGL norm, rather than just the group component.

**GAP Safe**  An exact feature reduction method for SGL was proposed in Ndiaye et al. (2016a) under linear regression. The approach makes use of the subdifferential inclusion equation of Fermat's rule (Bauschke & Combettes, 2017):

$$\mathbf{X}^\top \hat{\Theta}^{(\lambda, \|\cdot\|_{\text{sgl}})} \in \partial \| \cdot \|_{\text{sgl}}(\hat{\beta}^{(\lambda, \|\cdot\|_{\text{sgl}})}),$$

where $\hat{\Theta}$ is the solution to the dual formulation of Equation 1. Using this, exact (theoretical) rules are derived to determine which variables and groups are inactive at the optimal solution. The rules are theoretical as they rely on $\hat{\Theta}^{\lambda, \|\cdot\|_{\text{sgl}}}$, which is not available in practice. Instead, a safe region is constructed that contains the optimal dual solution; in Ndiaye et al. (2016a) it is taken as a sphere, but other regions can also be used (such as domes). Due to the strict requirements on these safe regions, the reduction is generally more conservative.

The safe sphere is defined as $B(\Theta_c, r)$ with center $\Theta_c$ and radius $r$. An ideal region would be such that $r$ is small and the center is close to $\hat{\Theta}^{\lambda, \|\cdot\|_{\text{sgl}}}$. Using this safe region, the GAP safe rules at $\lambda_{k+1}$ are derived as, for a variable $i$ and group $g$,

$$\text{Variable screening: } |X_i^\top \Theta_c| + r\|X_i\|_2 < \tau \implies \hat{\beta}_i(\lambda_{k+1}) = 0.$$
$$\text{Group screening: } \mathcal{T}_g < (1-\alpha)\sqrt{p_g} \implies \hat{\beta}^{(g)}(\lambda_{k+1}) \equiv \mathbf{0},$$

where

$$\mathcal{T}_g = \begin{cases} \|S(X_g^\top \Theta_c, \alpha)\| + r\|X_g\|, & \text{if } \|X_g^\top \Theta_c\|_\infty > \alpha, \\ (\|X_g^\top \Theta_c\|_\infty + r\|X_g\| - \alpha)_+, & \text{otherwise.} \end{cases}$$

The center $\Theta_c$ and the radius $r$ are derived using the duality gap and are calculated at iteration $t$ in an iterative algorithm as

$$\Theta_t(\beta_{(t)}) = \frac{y - \mathbf{X}\beta_{(t)}}{\max(\lambda_{k+1}, \|X^\top(y - \mathbf{X}\beta_{(t)})\|_{\text{sgl}}^*)}, \quad r_t(\beta_{(t)}, \Theta_t) = \sqrt{\frac{2P_{\lambda_{k+1}, \alpha}(\beta_{(t)}) - D_{\lambda_{k+1}}(\Theta_t)}{\lambda_{k+1}^2}},$$

where $P_{\lambda,\alpha}$ and $D_\lambda$ are the primal and dual objectives, and $\beta_{(t)}$ is the primal value at iteration $t$. The radius and center are expensive to evaluate, so are calculated only every 10 iterations (Ndiaye et al., 2016a).

The above formulation combines both dynamic and sequential screening. The method can also be implemented using just sequential screening, in which the primal values used in the calculation of the center and radius are from $\lambda_k$.

For both the GAP safe rules and DFR, theoretically it would be possible to exactly identify the active sets, but both instead require approximations. While GAP safe has different implementations, we present the best performing versions in our studies.

## D. Synthetic Data Analysis

This section complements Section 4.1 by providing further information about the simulation set-up and additional results for the synthetic data. Additional tables and figures are provided that further showcase the effectiveness of DFR, including under a logistic model (Appendix D.6).

*Notes:* The GAP safe methods are only applicable under linear regression and caused computational issues under uneven groups. Given their poor performance in the simulations considered (Figures 1, 2, and 3), they were excluded from the remaining simulations. Additionally, note that an intercept was only applied for linear models. Applying an intercept centers the response data, which is not applicable to a binary response. As we are interested in computational cost, not predictive performance, adding an intercept to logistic models provides no benefit.

### D.1. Metrics

The following metrics are shown in the tables in the Appendix:

- $\mathcal{A}_v, \mathcal{A}_g$: the number of active variables/groups.

- $\mathcal{C}_v, \mathcal{C}_g$: the number of variables/groups in the candidate sets.

- $\mathcal{O}_v, \mathcal{O}_g$: the number of variables/groups used in the optimization process. As per Algorithms A1 and A2, $\mathcal{O}_v = \mathcal{C}_v \cup \mathcal{A}_v$. However, $\mathcal{O}_g$ is not produced as $\mathcal{O}_g = \mathcal{C}_g \cup \mathcal{A}_g$. Instead, $\mathcal{O}_g$ are the groups for which there are variables present in $\mathcal{O}_v$ to give a measure of the number groups used in the optimization.

- $\mathcal{K}_v, \mathcal{K}_g$: the number of variable/group KKT violations. DFR only checks for variable violations and sparsegl only checks for group violations.

- $\mathcal{O}_v / \mathcal{A}_v$ and $\mathcal{O}_g / \mathcal{A}_g$: the proportion of variables/groups used in the optimization against the number active. Defines how efficient the rules are. A low value is best.

- $\mathcal{O}_v / p$ and $\mathcal{O}_g / m$: the variable/group input proportion, as defined in Section 4. A low value is best.

- $\ell_2$ distance to no screen: $\ell_2$ distance from the fitted values obtained with screening to without.

- IF: the improvement factor, as defined in Section 4.

### D.2. Setup

Table A2: Default model, data, and algorithm parameters for the synthetic and real data analyses.

| Category | Parameter | Values | |
| --- | --- | --- | --- |
| | | **Synthetic** | **Real** |
| **Model** | | | |
| | $\alpha$ | 0.95 | 0.95 |
| | $b_1 = b_2$ (aSGL only) | 0.1 | 0.1 |
| | Path length ($l$) | 50 | 100 |
| | Path termination ($\lambda_l$) | $0.1\lambda_1$ | $0.2\lambda_1$ |
| | Path shape | Log-linear | Log-linear |
| **Data** | | | |
| | $p$ | 1000 | - |
| | $n$ | 200 | - |
| | $m$ (uneven cases) | 22 | - |
| | $m$ (even cases) | 50 | - |
| | Group sizes (uneven cases) | $[3, 100]$ | - |
| | Group sizes (even cases) | 20 | - |
| | Signal $\beta$ (signal strength of zero) | $\mathcal{N}(0, 4)$ | - |
| | Variable sparsity | 0.2 | - |
| | Group sparsity | 0.2 | - |
| | Correlation ($\rho$) | 0.3 | - |
| | Noise ($\epsilon$) | $\mathcal{N}(0, 1)$ | - |
| **Algorithm (ATOS/BCD)** | | | |
| | Maximum iterations | 5000 | 10000 |
| | Backtracking (ATOS only) | 0.7 | 0.7 |
| | Maximum backtracking iterations (ATOS only) | 100 | 100 |
| | Convergence tolerance | $10^{-5}$ | $10^{-5}$ |
| | Standardization | $\ell_2$ | $\ell_2$ |
| | Intercept | Yes for linear | Yes for linear |
| | Warm starts | Yes | Yes |

## D.3. Runtime Breakdown

The runtime breakdowns for two cases are presented to illustrate the computational cost of screening.

Table A3: Figure 1 runtime breakdown.

| Component | DFR-SGL | DFR-aSGL |
|---|---|---|
| Fitting algorithm | 88% | 86% |
| $\epsilon$-norm evaluation | 3.9% | 3.6% |
| Group screening | 3.9% | 3.6% |
| Variable screening | 0.01% | 0.01% |
| KKT checks | 0.6% | 0.6% |

Table A4: *scheetz* runtime breakdown (Figure 8).

| Component | DFR-SGL | DFR-aSGL |
|---|---|---|
| Fitting algorithm | 77% | 65% |
| $\epsilon$-norm evaluation | 0.46% | 0.57% |
| Group screening | 0.46% | 0.58% |
| Variable screening | 0.01% | 0.02% |
| KKT checks | 0.2% | 0.3% |

## D.4. Additional Results for the Linear Model

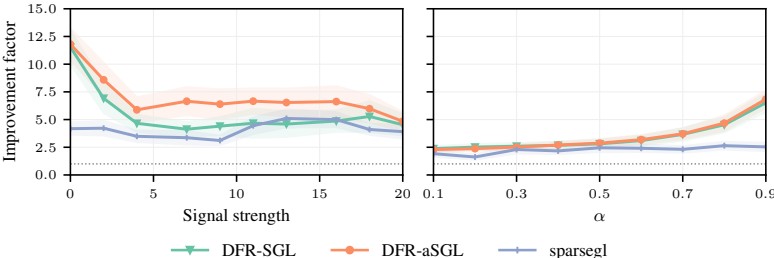

Figure A2: The improvement factor for the strong rules applied to synthetic data, under the linear model, as a function of the signal strength (left) and $\alpha$ (right), with $95\%$ confidence intervals.

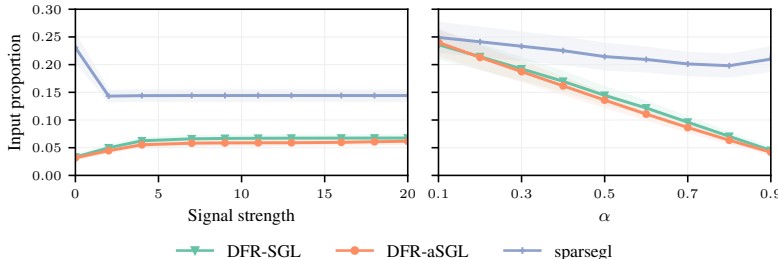

Figure A3: The input proportion for the strong rules applied to synthetic data, under the linear model, as a function of the signal strength (left) and $\alpha$ (right), with $95\%$ confidence intervals.

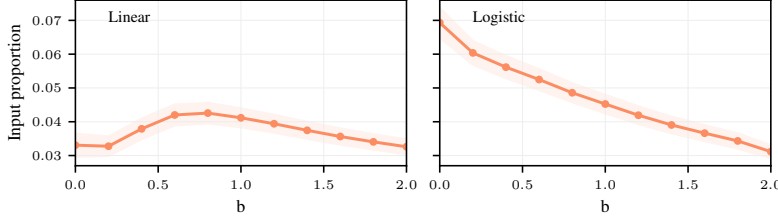

Figure A4: The input proportion of DFR-aSGL under different weights $b_1 = b_2$, shown for the linear (left) and logistic (right) models, with $95\%$ confidence intervals.

### D.4.1. TABLES FOR GAP SAFE SIMULATIONS

Table A5: Group screening metrics corresponding to the GAP safe simulations (Figures 1, 2, and 3) averaged over all cases and path points, shown with standard errors.

| | CARDINALITY | | | | INPUT PROPORTION | |
| METHOD | $\mathcal{A}_g$ | $\mathcal{C}_g$ | $\mathcal{O}_g$ | $\mathcal{K}_g$ | $\mathcal{O}_g / \mathcal{A}_g$ | $\mathcal{O}_g / m$ |
|---|---|---|---|---|---|---|
| DFR-ASGL | $7.86 \pm 0.02$ | $10.00 \pm 0.03$ | $10.00 \pm 0.03$ | – | $1.1751 \pm 7 \times 10^{-4}$ | $0.2001 \pm 5 \times 10^{-4}$ |
| DFR-SGL | $8.16 \pm 0.02$ | $10.47 \pm 0.02$ | $10.47 \pm 0.02$ | – | $1.1937 \pm 8 \times 10^{-4}$ | $0.2094 \pm 5 \times 10^{-4}$ |
| DFR-SGL BCD | $8.45 \pm 0.02$ | $10.07 \pm 0.02$ | $10.07 \pm 0.02$ | – | $1.1893 \pm 8 \times 10^{-4}$ | $0.2015 \pm 4 \times 10^{-4}$ |
| SPARSEGL | $8.16 \pm 0.02$ | $11.72 \pm 0.04$ | $11.72 \pm 0.04$ | $3 \times 10^{-5} \pm 2 \times 10^{-5}$ | $1.5837 \pm 0.0111$ | $0.2344 \pm 7 \times 10^{-4}$ |
| GAP SEQUENTIAL | $8.55 \pm 0.02$ | $10.20 \pm 0.02$ | $10.20 \pm 0.02$ | – | $1.1934 \pm 0.0018$ | $0.2040 \pm 4 \times 10^{-4}$ |
| GAP DYNAMIC | $8.55 \pm 0.02$ | $10.20 \pm 0.02$ | $10.20 \pm 0.02$ | – | $1.1917 \pm 0.0017$ | $0.2039 \pm 4 \times 10^{-4}$ |

Table A6: Variable screening metrics corresponding to the GAP safe simulations (Figures 1, 2, and 3) averaged over all cases and path points, shown with standard errors.

| | CARDINALITY | | | | INPUT PROPORTION | |
| METHOD | $\mathcal{A}_v$ | $\mathcal{C}_v$ | $\mathcal{O}_v$ | $\mathcal{K}_v$ | $\mathcal{O}_v / \mathcal{A}_v$ | $\mathcal{O}_v / p$ |
|---|---|---|---|---|---|---|
| DFR-ASGL | $61.77 \pm 0.22$ | $36.70 \pm 0.34$ | $95.99 \pm 0.42$ | $0.0092 \pm 3 \times 10^{-4}$ | $1.3464 \pm 0.0011$ | $0.0960 \pm 4 \times 10^{-4}$ |
| DFR-SGL | $65.37 \pm 0.22$ | $38.30 \pm 0.34$ | $101.07 \pm 0.43$ | $0 \pm 0$ | $1.3576 \pm 0.0011$ | $0.1011 \pm 4 \times 10^{-4}$ |
| DFR-SGL BCD | $72.69 \pm 0.25$ | $23.35 \pm 0.09$ | $93.11 \pm 0.32$ | $0.0250 \pm 0.0054$ | $1.3281 \pm 0.0011$ | $0.0931 \pm 3 \times 10^{-4}$ |
| SPARSEGL | $65.41 \pm 0.22$ | $234.39 \pm 0.70$ | $234.39 \pm 0.70$ | – | $11.5604 \pm 0.2001$ | $0.2344 \pm 7 \times 10^{-4}$ |
| GAP SEQUENTIAL | $73.01 \pm 0.25$ | $93.85 \pm 0.33$ | $93.85 \pm 0.33$ | – | $1.3292 \pm 0.0022$ | $0.0938 \pm 3 \times 10^{-4}$ |
| GAP DYNAMIC | $73.01 \pm 0.25$ | $93.38 \pm 0.33$ | $93.38 \pm 0.33$ | – | $1.2927 \pm 0.0020$ | $0.0934 \pm 3 \times 10^{-4}$ |

Table A7: Model fitting metrics corresponding to the GAP safe simulations (Figures 1, 2, and 3) averaged over all cases and path points, shown with standard errors. The timing results are the average time taken to evaluate the full path on a dataset.

| | TIMINGS | | | ITERATIONS | | $\ell_2$ DISTANCE | FAILED CONVERGENCE | |
| METHOD | NO SCREEN (S) | SCREEN (S) | IF | NO SCREEN | SCREEN | TO NO SCREEN | NO SCREEN | SCREEN |
|---|---|---|---|---|---|---|---|---|
| DFR-ASGL | $659.89 \pm 6.50$ | $154.92 \pm 2.39$ | $7.01 \pm 0.15$ | $271.60 \pm 2.36$ | $174.55 \pm 2.49$ | $4 \times 10^{-4} \pm 3 \times 10^{-6}$ | $0 \pm 0$ | $0 \pm 0$ |
| DFR-SGL | $685.49 \pm 6.27$ | $157.1 \pm 2.24$ | $7.98 \pm 0.20$ | $286.13 \pm 2.04$ | $185.47 \pm 2.43$ | $4 \times 10^{-4} \pm 3 \times 10^{-6}$ | $0 \pm 0$ | $0 \pm 0$ |
| DFR-SGL BCD | $131.46 \pm 1.12$ | $32.02 \pm 0.55$ | $5.55 \pm 0.08$ | $252.26 \pm 1.94$ | $139.39 \pm 1.15$ | $2 \times 10^{-7} \pm 1 \times 10^{-8}$ | $0 \pm 0$ | $0 \pm 0$ |
| SPARSEGL | $685.49 \pm 6.27$ | $275.37 \pm 4.43$ | $3.44 \pm 0.08$ | $286.13 \pm 2.04$ | $278.18 \pm 2.28$ | $4 \times 10^{-4} \pm 3 \times 10^{-6}$ | $0 \pm 0$ | $0 \pm 0$ |
| GAP SEQUENTIAL | $0.11 \pm 3 \times 10^{-3}$ | $0.11 \pm 3 \times 10^{-3}$ | $0.98 \pm 0.01$ | – | – | – | – | |
| GAP DYNAMIC | $0.11 \pm 3 \times 10^{-3}$ | $0.11 \pm 3 \times 10^{-3}$ | $1.00 \pm 0.01$ | – | – | – | – | – |

### D.4.2. TABLES FOR OTHER SIMULATIONS

Table A8: Group screening metrics corresponding to the other linear model simulations (Figures 4, 5, 6, A2, and A3 and Table 1) averaged over all cases and path points, shown with standard errors.

| | CARDINALITY | | | | INPUT PROPORTION | |
| METHOD | $\mathcal{A}_g$ | $\mathcal{C}_g$ | $\mathcal{O}_g$ | $\mathcal{K}_g$ | $\mathcal{O}_g / \mathcal{A}_g$ | $\mathcal{O}_g / m$ |
|---|---|---|---|---|---|---|
| DFR-ASGL | $8.46 \pm 0.03$ | $10.55 \pm 0.04$ | $10.54 \pm 0.04$ | – | $1.1414 \pm 5 \times 10^{-4}$ | $0.1968 \pm 4 \times 10^{-4}$ |
| DFR-SGL | $8.51 \pm 0.03$ | $11.16 \pm 0.04$ | $11.16 \pm 0.04$ | – | $1.1792 \pm 6 \times 10^{-4}$ | $0.2084 \pm 4 \times 10^{-4}$ |
| SPARSEGL | $8.51 \pm 0.03$ | $10.25 \pm 0.04$ | $10.25 \pm 0.04$ | $8 \times 10^{-5} \pm 2 \times 10^{-5}$ | $1.2191 \pm 0.0026$ | $0.2083 \pm 4 \times 10^{-4}$ |

Table A9: Varible screening metrics corresponding to the other linear model simulations (Figures 4, 5, 6, A2, and A3 and Table 1) averaged over all cases and path points, shown with standard errors.

| | CARDINALITY | | | | INPUT PROPORTION | |
|---|---|---|---|---|---|---|
| METHOD | $\mathcal{A}_v$ | $\mathcal{C}_v$ | $\mathcal{O}_v$ | $\mathcal{K}_v$ | $\mathcal{O}_v \,/\, \mathcal{A}_v$ | $\mathcal{O}_v \,/\, p$ |
| DFR-ASGL | $50.95 \pm 0.13$ | $29.90 \pm 0.15$ | $78.92 \pm 0.22$ | $0.0297 \pm 4 \times 10^{-4}$ | $1.4805 \pm 0.0011$ | $0.0634 \pm 2 \times 10^{-4}$ |
| DFR-SGL | $54.40 \pm 0.13$ | $32.78 \pm 0.15$ | $85.16 \pm 0.22$ | $4 \times 10^{-6} \pm 4 \times 10^{-6}$ | $1.5057 \pm 0.0014$ | $0.0676 \pm 2 \times 10^{-4}$ |
| SPARSEGL | $54.42 \pm 0.13$ | $331.75 \pm 0.86$ | $331.75 \pm 0.86$ | $-$ | $11.4845 \pm 0.0882$ | $0.2205 \pm 5 \times 10^{-4}$ |

Table A10: Model fitting metrics corresponding to the other linear model simulations (Figures 4, 5, 6, A2, and A3 and Table 1) averaged over all cases and path points, shown with standard errors. The timing results are the average time taken to evaluate the full path on a dataset.

| | TIMINGS | | | ITERATIONS | | $\ell_2$ DISTANCE | FAILED CONVERGENCE | |
|---|---|---|---|---|---|---|---|---|
| METHOD | NO SCREEN (S) | SCREEN (S) | IF | NO SCREEN | SCREEN | TO NO SCREEN | NO SCREEN | SCREEN |
| DFR-ASGL | $723.99 \pm 34.04$ | $77.00 \pm 2.23$ | $11.19 \pm 0.41$ | $415.87 \pm 7.41$ | $227.93 \pm 3.51$ | $7 \times 10^{-5} \pm 5 \times 10^{-7}$ | $0 \pm 0$ | $0 \pm 0$ |
| DFR-SGL | $251.59 \pm 3.93$ | $51.73 \pm 0.89$ | $9.15 \pm 0.15$ | $413.10 \pm 7.12$ | $240.78 \pm 3.78$ | $6 \times 10^{-5} \pm 5 \times 10^{-7}$ | $0 \pm 0$ | $0 \pm 0$ |
| SPARSEGL | $251.59 \pm 3.93$ | $138.18 \pm 5.62$ | $3.59 \pm 0.05$ | $413.10 \pm 7.12$ | $353.66 \pm 5.79$ | $6 \times 10^{-5} \pm 5 \times 10^{-7}$ | $0 \pm 0$ | $0 \pm 0$ |

### D.4.3. CROSS-VALIDATION

Table A11: The improvement factor for the strong rules applied to synthetic data, under the linear and logistic models, with 10-fold CV, with standard errors.

| Method | Linear | Logistic |
|---|---|---|
| DFR-aSGL | $3.9 \pm 0.2$ | $2.3 \pm 0.1$ |
| DFR-SGL | $4.2 \pm 0.3$ | $2.6 \pm 0.1$ |
| sparsegl | $2.0 \pm 0.2$ | $2.1 \pm 0.1$ |

## D.5. Interaction Models

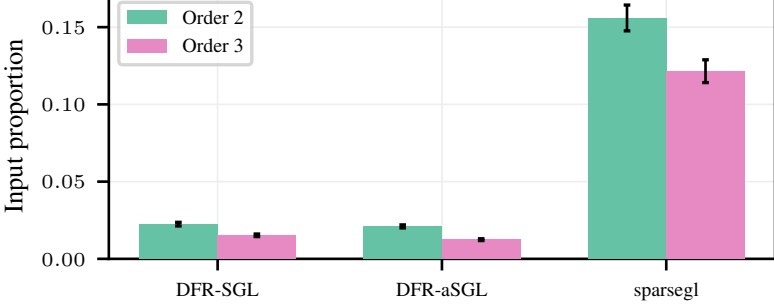

Figure A5: The input proportion for the strong rules applied to synthetic interaction data, under the linear model, with standard errors. The parameters of the data were set as $p = 400, n = 80$, and $m = 52$ groups of sizes in $[3, 15]$. The interaction input dimensionality was $p_{O_2} = 2111$ and $p_{O_3} = 7338$, with no interaction hierarchy imposed. The sparsity proportion of interaction variables was set to $0.3$ (with the same signal as the marginal effects).

### D.6. Results for the Logistic Model

The data input components $\mathbf{X}$, $\beta$, and $\epsilon$ for the logistic model were generated as for the linear models. The class probabilities for the response were calculated using $\sigma(\mathbf{X}\beta + \epsilon)$, where $\sigma$ is the sigmoid function.

Table A12: The improvement factor for the strong rules applied to synthetic interaction data, under the logistic model, with standard errors. The parameters of the data were set as $p = 400, n = 80$, and $m = 52$ groups of sizes in $[3, 15]$. The interaction input dimensionality was $p_{O_2} = 2111$ and $p_{O_3} = 7338$, with no interaction hierarchy imposed. The sparsity proportion of interaction variables was set to 0.3 (with the same signal as the marginal effects).

| | Interaction | |
| Method | Order 2 | Order 3 |
| --- | --- | --- |
| DFR-aSGL | $6.7 \pm 0.4$ | $12.2 \pm 0.4$ |
| DFR-SGL | $5.8 \pm 0.2$ | $8.3 \pm 0.4$ |
| sparsegl | $1.0 \pm 0.1$ | $2.1 \pm 0.3$ |

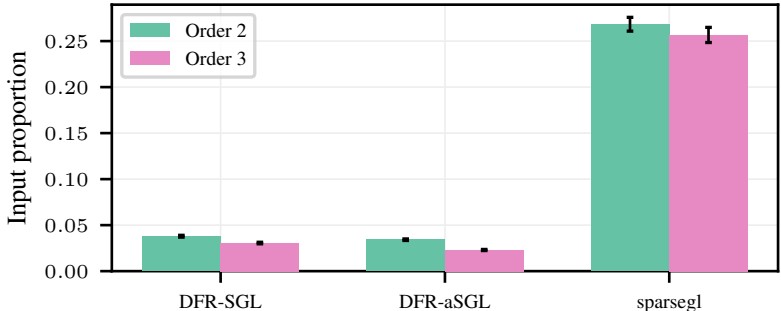

Figure A6: The input proportion for the strong rules applied to synthetic interaction data, under the logistic model, with standard errors. The parameters of the data were set as $p = 400, n = 80$, and $m = 52$ groups of sizes in $[3, 15]$. The interaction input dimensionality was $p_{O_2} = 2111$ and $p_{O_3} = 7338$, with no interaction hierarchy imposed. The sparsity proportion of interaction variables was set to 0.3 (with the same signal as the marginal effects).

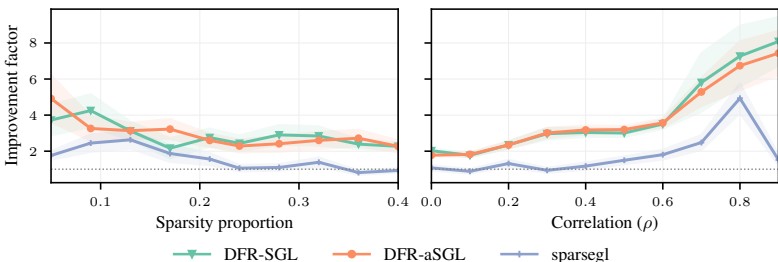

Figure A7: The improvement factor for the strong rules applied to synthetic data, under the logistic model, as a function of the sparsity proportion (left) and data correlation (right), with $95\%$ confidence intervals.

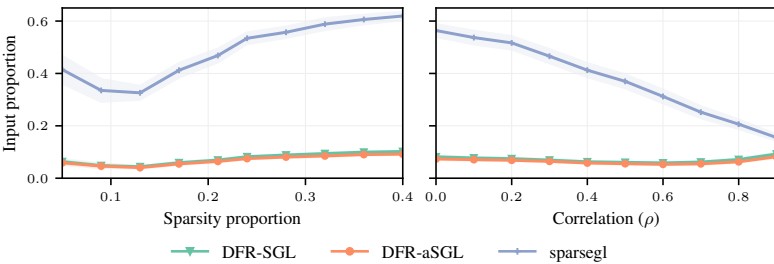

Figure A8: The input proportion for the strong rules applied to synthetic data, under the logistic model, as a function of the sparsity proportion (left) and data correlation (right), with 95% confidence intervals.

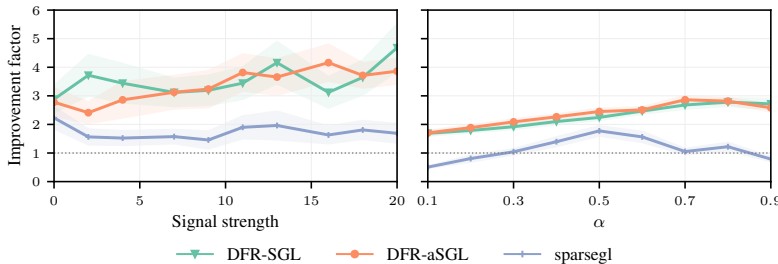

Figure A9: The improvement factor for the strong rules applied to synthetic data, under the logistic model, as a function of the signal strength (left) and $\alpha$ (right), with 95% confidence intervals.

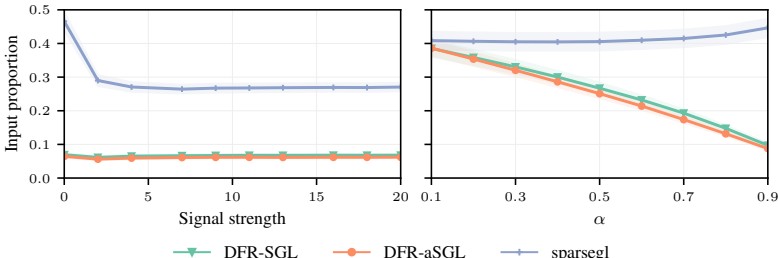

Figure A10: The input proportion for the strong rules applied to synthetic data, under the logistic model, as a function of the signal strength (left) and $\alpha$ (right), with 95% confidence intervals.

### D.6.1. TABLES FOR LOGISTIC SIMULATIONS

Table A13: Group screening metrics corresponding to the logistic model simulations (Figures A7, A8, A9, and A10 and Table A12) averaged over all cases and path points, shown with standard errors.

| METHOD | CARDINALITY | | | | INPUT PROPORTION | |
|---|---|---|---|---|---|---|
| | $\mathcal{A}_g$ | $\mathcal{C}_g$ | $\mathcal{O}_g$ | $\mathcal{K}_g$ | $\mathcal{O}_g / \mathcal{A}_g$ | $\mathcal{O}_g / m$ |
| DFR-aSGL | $8.08 \pm 0.01$ | $8.55 \pm 0.02$ | $8.55 \pm 0.02$ | – | $1.0292 \pm 6 \times 10^{-4}$ | $0.3528 \pm 6 \times 10^{-4}$ |
| DFR-SGL | $8.56 \pm 0.02$ | $9.30 \pm 0.02$ | $9.30 \pm 0.02$ | – | $1.0554 \pm 7 \times 10^{-4}$ | $0.3823 \pm 7 \times 10^{-4}$ |
| SPARSEGL | $8.56 \pm 0.02$ | $8.87 \pm 0.02$ | $8.87 \pm 0.02$ | $5 \times 10^{-5} \pm 2 \times 10^{-5}$ | $1.0808 \pm 0.0029$ | $0.3697 \pm 7 \times 10^{-4}$ |

Table A14: Variable screening metrics corresponding to the logistic model simulations (Figures A7, A8, A9, and A10 and Table A12) averaged over all cases and path points, shown with standard errors.

| METHOD | CARDINALITY | | | | INPUT PROPORTION | |
|---|---|---|---|---|---|---|
| | $\mathcal{A}_v$ | $\mathcal{C}_v$ | $\mathcal{O}_v$ | $\mathcal{K}_v$ | $\mathcal{O}_v / \mathcal{A}_v$ | $\mathcal{O}_v / p$ |
| DFR-ASGL | $79.76 \pm 0.24$ | $35.27 \pm 0.10$ | $111.79 \pm 0.32$ | $0.0155 \pm 3 \times 10^{-4}$ | $1.5263 \pm 0.0012$ | $0.1075 \pm 3 \times 10^{-4}$ |
| DFR-SGL | $84.23 \pm 0.25$ | $40.13 \pm 0.12$ | $120.84 \pm 0.34$ | $9 \times 10^{-6} \pm 7 \times 10^{-6}$ | $1.5503 \pm 0.0013$ | $0.1154 \pm 3 \times 10^{-4}$ |
| SPARSEGL | $84.26 \pm 0.25$ | $445.21 \pm 1.05$ | $445.21 \pm 1.05$ | $-$ | $11.6349 \pm 0.1015$ | $0.4004 \pm 7 \times 10^{-4}$ |

Table A15: Model fitting metrics corresponding to the logistic model simulations (Figures A7, A8, A9, and A10 and Table A12) averaged over all cases and path points, shown with standard errors. The timing results are the average time taken to evaluate the full path on a dataset.

| METHOD | TIMINGS | | | ITERATIONS | | $\ell_2$ DISTANCE | FAILED CONVERGENCE | |
|---|---|---|---|---|---|---|---|---|
| | NO SCREEN (S) | SCREEN (S) | IF | NO SCREEN | SCREEN | TO NO SCREEN | NO SCREEN | SCREEN |
| DFR-ASGL | $132.10 \pm 7.21$ | $34.75 \pm 0.66$ | $3.37 \pm 0.05$ | $125.73 \pm 2.77$ | $78.95 \pm 1.10$ | $9 \times 10^{-10} \pm 2 \times 10^{-11}$ | $0 \pm 0$ | $0 \pm 0$ |
| DFR-SGL | $103.16 \pm 3.37$ | $34.27 \pm 0.56$ | $3.43 \pm 0.05$ | $141.42 \pm 2.92$ | $87.88 \pm 1.30$ | $2 \times 10^{-10} \pm 8 \times 10^{-12}$ | $0 \pm 0$ | $0 \pm 0$ |
| SPARSEGL | $103.16 \pm 3.37$ | $109.93 \pm 4.69$ | $1.51 \pm 0.03$ | $141.42 \pm 2.92$ | $121.29 \pm 2.11$ | $2 \times 10^{-10} \pm 3 \times 10^{-12}$ | $0 \pm 0$ | $0 \pm 0$ |

# E. Real Data Analysis

## E.1. Data Description

- brca1: Gene expression data for breast cancer tissue samples.

    - Response (continuous): Gene expression measurements for the BRCA1 gene.
    - Data matrix: Gene expression measurements for the other genes.
    - Grouping structure: Variables are grouped via singular value decomposition.

- scheetz: Gene expression data in the mammalian eye.

    - Response (continuous): Gene expression measurements for the Trim32 gene.
    - Data matrix: Gene expression measurements for the other genes.
    - Grouping structure: Variables are grouped via singular value decomposition.

- trust-experts: Survey response data as to how much participants trust experts (e.g., doctors, nurses, scientists) to provide COVID-19 news and information.

    - Response (continuous): The trust level of each participant.
    - Data matrix: Contingency table including factors about participants (e.g., age, gender, ethnicity).
    - Grouping structure: The factor levels grouped into their original factors.

- adenoma: Transcriptome profile data to identify the formation of colorectal adenomas, which are the predominate cause of colorectal cancers.

    - Response (binary): Labels classifying whether the sample came from an adenoma or normal mucosa.
    - Data matrix: Transcriptome profile measurements.
    - Grouping structure: Genes were assigned to pathways from all nine gene sets on the Molecular Signature Database.[1]

- celiac: Gene expression data of primary leucocytes to identify celiac disease.

    - Response (binary): Labels classifying patients into whether they have celiac disease.
    - Data matrix: Gene expression measurements from the primary leucocytes.
    - Grouping structure: Genes were mapped to pathways from all nine Molecular Signature Database gene sets.[1]

- tumour: Gene expression data of pancreative cancer samples to identify tumorous tissue.

    - Response (binary): Labels classifying whether samples are from normal or tumour tissue.
    - Data matrix: Gene expression measurements.
    - Grouping structure: Genes were mapped to pathways from all nine Molecular Signature Database gene sets.[1]

Table A16: Dataset information for the six datasets used in the real data analysis.

| Dataset | $p$ | $n$ | $m$ | Group sizes | Type | Source |
|---|---|---|---|---|---|---|
| brca1 | 17322 | 536 | 243 | $[1, 6505]$ | Linear | (National Cancer Institute, 1988)[2] |
| scheetz | 18975 | 120 | 85 | $[1, 6274]$ | Linear | (Scheetz et al., 2006)[2] |
| trust-experts | 101 | 9759 | 7 | $[4, 51]$ | Linear | (Salomon et al., 2021)[3] |
| adenoma | 18559 | 64 | 313 | $[1, 741]$ | Logistic | (Sabates-Bellver et al., 2007)[4] |
| celiac | 14657 | 132 | 276 | $[1, 617]$ | Logistic | (Heap et al., 2009)[4] |
| tumour | 18559 | 52 | 313 | $[1, 741]$ | Logistic | (Pei et al., 2009; Ellsworth et al., 2013; Li et al., 2016)[4] |

## E.2. Additional Results for the Real Data

---

[1] downloaded on 08/2024 from gsea-msigdb.org/gsea/msigdb/human/collections.jsp.
[2] downloaded on 08/2024 from https://iowabiostat.github.io/data-sets/.
[3] downloaded on 08/2024 from https://github.com/dajmcdon/sparsegl.
[4] downloaded on 08/2024 from https://www.ncbi.nlm.nih.gov/.

Table A17: Group screening metrics corresponding to the real data studies (Figures 8, 9, and 10) averaged over all path points, shown with standard errors.

| METHOD | DATASET | CARDINALITY | | | | INPUT PROPORTION | |
| --- | --- | --- | --- | --- | --- | --- | --- |
| | | $\mathcal{A}_g$ | $\mathcal{C}_g$ | $\mathcal{O}_g$ | $\mathcal{K}_g$ | $\mathcal{O}_g / \mathcal{A}_g$ | $\mathcal{O}_g / m$ |
| DFR-ASGL | ADENOMA | $1.41 \pm 0.08$ | $1.38 \pm 0.08$ | $1.42 \pm 0.08$ | - | $1.0053 \pm 0.0053$ | $0.0046 \pm 3 \times 10^{-4}$ |
| DFR-SGL | ADENOMA | $3.30 \pm 0.21$ | $13.46 \pm 0.68$ | $13.46 \pm 0.68$ | - | $4.6121 \pm 0.2003$ | $0.0430 \pm 0.0022$ |
| SPARSEGL | ADENOMA | $3.32 \pm 0.21$ | $8.59 \pm 0.53$ | $8.59 \pm 0.53$ | $0 \pm 0$ | $2.6141 \pm 0.0942$ | $0.0274 \pm 0.0017$ |
| DFR-ASGL | CELIAC | $19.41 \pm 1.65$ | $29.94 \pm 2.45$ | $28.82 \pm 2.35$ | - | $1.4412 \pm 0.0273$ | $0.1044 \pm 0.0085$ |
| DFR-SGL | CELIAC | $15.35 \pm 1.45$ | $22.04 \pm 2.03$ | $22.04 \pm 2.03$ | - | $1.4367 \pm 0.0276$ | $0.0799 \pm 0.0074$ |
| SPARSEGL | CELIAC | $15.36 \pm 1.46$ | $19.32 \pm 1.84$ | $19.32 \pm 1.84$ | $0 \pm 0$ | $1.2415 \pm 0.0213$ | $0.0700 \pm 0.0067$ |
| DFR-ASGL | BRCA1 | $3.84 \pm 0.26$ | $4.17 \pm 0.33$ | $4.16 \pm 0.33$ | - | $1.0439 \pm 0.0121$ | $0.0171 \pm 0.0013$ |
| DFR-SGL | BRCA1 | $5.60 \pm 0.48$ | $6.90 \pm 0.60$ | $6.90 \pm 0.60$ | - | $1.2023 \pm 0.0218$ | $0.0284 \pm 0.0025$ |
| SPARSEGL | BRCA1 | $5.59 \pm 0.48$ | $6.25 \pm 0.55$ | $6.25 \pm 0.55$ | $0 \pm 0$ | $1.1062 \pm 0.0176$ | $0.0257 \pm 0.0022$ |
| DFR-ASGL | SCHEETZ | $2.19 \pm 0.12$ | $2.37 \pm 0.16$ | $2.39 \pm 0.16$ | - | $1.0515 \pm 0.0121$ | $0.0282 \pm 0.0019$ |
| DFR-SGL | SCHEETZ | $0.61 \pm 0.08$ | $0.86 \pm 0.12$ | $0.86 \pm 0.12$ | - | $1.3537 \pm 0.0531$ | $0.0101 \pm 0.0014$ |
| SPARSEGL | SCHEETZ | $0.61 \pm 0.08$ | $0.73 \pm 0.10$ | $0.73 \pm 0.10$ | $0 \pm 0$ | $1.1829 \pm 0.0486$ | $0.0086 \pm 0.0012$ |
| DFR-ASGL | TRUST-EXPERTS | $3.41 \pm 0.08$ | $3.37 \pm 0.09$ | $3.41 \pm 0.08$ | - | $1.0000 \pm 0.0000$ | $0.4877 \pm 0.0118$ |
| DFR-SGL | TRUST-EXPERTS | $3.34 \pm 0.08$ | $3.37 \pm 0.08$ | $3.37 \pm 0.08$ | - | $1.0185 \pm 0.0117$ | $0.4820 \pm 0.0113$ |
| SPARSEGL | TRUST-EXPERTS | $3.30 \pm 0.09$ | $0.04 \pm 0.02$ | $3.30 \pm 0.09$ | $0 \pm 0$ | $1.0000 \pm 0.0000$ | $0.4719 \pm 0.0127$ |
| DFR-ASGL | TUMOUR | $3.94 \pm 0.22$ | $4.96 \pm 0.30$ | $4.98 \pm 0.30$ | - | $1.2228 \pm 0.0240$ | $0.0159 \pm 1 \times 10^{-3}$ |
| DFR-SGL | TUMOUR | $5.02 \pm 0.24$ | $8.80 \pm 0.38$ | $8.80 \pm 0.38$ | - | $1.8253 \pm 0.0357$ | $0.0281 \pm 0.0012$ |
| SPARSEGL | TUMOUR | $5.02 \pm 0.24$ | $6.77 \pm 0.29$ | $6.77 \pm 0.29$ | $0 \pm 0$ | $1.4276 \pm 0.0351$ | $0.0216 \pm 9 \times 10^{-4}$ |

Table A18: Variable screening metrics corresponding to the real data studies (Figures 8, 9, and 10) averaged over all path points, shown with standard errors.

| METHOD | DATASET | CARDINALITY | | | | INPUT PROPORTION | |
|---|---|---|---|---|---|---|---|
| | | $\mathcal{A}_v$ | $\mathcal{C}_v$ | $\mathcal{O}_v$ | $\mathcal{K}_v$ | $\mathcal{O}_v / \mathcal{A}_v$ | $\mathcal{O}_v / p$ |
| DFR-ASGL | ADENOMA | $3.38 \pm 0.21$ | $0.81 \pm 0.11$ | $4.15 \pm 0.30$ | $0.0404 \pm 0.0199$ | $1.1828 \pm 0.0259$ | $2 \times 10^{-4} \pm 2 \times 10^{-5}$ |
| DFR-SGL | ADENOMA | $14.38 \pm 1.11$ | $61.47 \pm 2.90$ | $75.51 \pm 3.85$ | $0 \pm 0$ | $8.9655 \pm 0.7776$ | $0.0041 \pm 2 \times 10^{-4}$ |
| SPARSEGL | ADENOMA | $14.41 \pm 1.12$ | $308.64 \pm 20.81$ | $308.64 \pm 20.81$ | - | $26.0099 \pm 1.6361$ | $0.0166 \pm 0.0011$ |
| DFR-ASGL | CELIAC | $40.13 \pm 3.85$ | $29.63 \pm 2.79$ | $68.61 \pm 6.51$ | $0.1010 \pm 0.0337$ | $1.6545 \pm 0.0442$ | $0.0047 \pm 4 \times 10^{-4}$ |
| DFR-SGL | CELIAC | $37.13 \pm 3.92$ | $26.11 \pm 3.05$ | $61.94 \pm 6.77$ | $0 \pm 0$ | $1.6187 \pm 0.0455$ | $0.0042 \pm 5 \times 10^{-4}$ |
| SPARSEGL | CELIAC | $37.26 \pm 3.94$ | $1019.13 \pm 106.78$ | $1019.13 \pm 106.78$ | - | $27.6753 \pm 1.1176$ | $0.0695 \pm 0.0073$ |
| DFR-ASGL | BRCA1 | $135.75 \pm 4.69$ | $31.93 \pm 2.77$ | $165.83 \pm 6.19$ | $0.0505 \pm 0.0221$ | $1.1998 \pm 0.0116$ | $0.0096 \pm 4 \times 10^{-4}$ |
| DFR-SGL | BRCA1 | $241.69 \pm 7.34$ | $63.42 \pm 4.26$ | $301.8 \pm 8.81$ | $0 \pm 0$ | $3.9386 \pm 2.6945$ | $0.0174 \pm 5 \times 10^{-4}$ |
| SPARSEGL | BRCA1 | $241.59 \pm 7.34$ | $6762.31 \pm 186.59$ | $6762.31 \pm 186.59$ | - | $95.8171 \pm 66.017$ | $0.3904 \pm 0.0108$ |
| DFR-ASGL | SCHEETZ | $743.43 \pm 13.50$ | $281.91 \pm 13.50$ | $1019.11 \pm 20.19$ | $0.0202 \pm 0.0142$ | $1.3678 \pm 0.0050$ | $0.0537 \pm 0.0011$ |
| DFR-SGL | SCHEETZ | $501.78 \pm 60.68$ | $344.92 \pm 50.24$ | $836.23 \pm 101.02$ | $0 \pm 0$ | $1.6688 \pm 0.0651$ | $0.0441 \pm 0.0053$ |
| SPARSEGL | SCHEETZ | $501.49 \pm 60.66$ | $3030.24 \pm 385.60$ | $3030.24 \pm 385.60$ | - | $6.1978 \pm 0.3472$ | $0.1597 \pm 0.0203$ |
| DFR-ASGL | TRUST-EXPERTS | $4.83 \pm 0.20$ | $0.17 \pm 0.04$ | $4.96 \pm 0.21$ | $0.0404 \pm 0.0199$ | $1.0231 \pm 0.0067$ | $0.0491 \pm 0.0021$ |
| DFR-SGL | TRUST-EXPERTS | $5.10 \pm 0.28$ | $0.35 \pm 0.07$ | $5.36 \pm 0.29$ | $0 \pm 0$ | $1.0608 \pm 0.0155$ | $0.0531 \pm 0.0029$ |
| SPARSEGL | TRUST-EXPERTS | $5.07 \pm 0.28$ | $0.27 \pm 0.14$ | $21.42 \pm 0.65$ | - | $4.9628 \pm 0.1448$ | $0.2121 \pm 0.0065$ |
| DFR-ASGL | TUMOUR | $7.40 \pm 0.58$ | $3.31 \pm 0.31$ | $10.57 \pm 0.84$ | $0.0202 \pm 0.0142$ | $1.3363 \pm 0.0394$ | $6 \times 10^{-4} \pm 5 \times 10^{-5}$ |
| DFR-SGL | TUMOUR | $10.70 \pm 0.72$ | $9.87 \pm 0.50$ | $20.34 \pm 1.09$ | $0 \pm 0$ | $2.3432 \pm 0.1189$ | $0.0011 \pm 6 \times 10^{-5}$ |
| SPARSEGL | TUMOUR | $10.70 \pm 0.72$ | $246.8 \pm 15.39$ | $246.8 \pm 15.39$ | - | $25.9945 \pm 0.9193$ | $0.0133 \pm 8 \times 10^{-4}$ |

Table A19: Model fitting metrics corresponding to the real data studies (Figures 8, 9, and 10) averaged over all path points, shown with standard errors. There are no standard errors for the timing results as the time to calculate the whole path was evaluated.

| METHOD | DATASET | TIMINGS | | | ITERATIONS | | $\ell_2$ DISTANCE | FAILED CONVERGENCE | |
| --- | --- | --- | --- | --- | --- | --- | --- | --- | --- |
| | | NO SCREEN (S) | SCREEN (S) | I.F. | NO SCREEN | SCREEN | TO NO SCREEN | NO SCREEN | SCREEN |
| DFR-ASGL | ADENOMA | 8476.04 | 7.91 | 1072.10 | $8903.76 \pm 270.64$ | $119.18 \pm 14.69$ | $7 \times 10^{-5} \pm 6 \times 10^{-6}$ | $0.8384 \pm 0.0372$ | $0 \pm 0$ |
| DFR-SGL | ADENOMA | 9017.70 | 149.10 | 60.48 | $9272.90 \pm 205.31$ | $4374.64 \pm 286.62$ | $3 \times 10^{-5} \pm 2 \times 10^{-6}$ | $0.8687 \pm 0.0341$ | $0 \pm 0$ |
| SPARSEGL | ADENOMA | 9017.70 | 198.36 | 45.46 | $9272.90 \pm 205.31$ | $5140.16 \pm 390.91$ | $3 \times 10^{-5} \pm 2 \times 10^{-6}$ | $0.8687 \pm 0.0341$ | $0.0404 \pm 0.0199$ |
| DFR-ASGL | CELIAC | 1188.14 | 15.86 | 74.89 | $1042.07 \pm 90.79$ | $120.96 \pm 14.51$ | $1 \times 10^{-6} \pm 2 \times 10^{-7}$ | $0 \pm 0$ | $0 \pm 0$ |
| DFR-SGL | CELIAC | 1391.78 | 10.31 | 134.95 | $1195.34 \pm 98.13$ | $75.37 \pm 9.60$ | $2 \times 10^{-7} \pm 1 \times 10^{-8}$ | $0 \pm 0$ | $0 \pm 0$ |
| SPARSEGL | CELIAC | 1391.78 | 16.49 | 84.40 | $1195.34 \pm 98.13$ | $93.29 \pm 8.86$ | $8 \times 10^{-8} \pm 3 \times 10^{-9}$ | $0 \pm 0$ | $0 \pm 0$ |
| DFR-ASGL | BRCA1 | 21889.33 | 119.16 | 183.69 | $1653.45 \pm 33.90$ | $243.27 \pm 11.70$ | $2 \times 10^{-9} \pm 7 \times 10^{-10}$ | $0 \pm 0$ | $0 \pm 0$ |
| DFR-SGL | BRCA1 | 22227.01 | 103.78 | 214.17 | $1674.07 \pm 19.77$ | $334.16 \pm 13.44$ | $2 \times 10^{-12} \pm 3 \times 10^{-13}$ | $0 \pm 0$ | $0 \pm 0$ |
| SPARSEGL | BRCA1 | 22227.01 | 4132.04 | 5.38 | $1674.07 \pm 19.77$ | $1580.73 \pm 44.23$ | $4 \times 10^{-14} \pm 1 \times 10^{-14}$ | $0 \pm 0$ | $0 \pm 0$ |
| DFR-ASGL | SCHEETZ | 90569.05 | 132.20 | 685.08 | $6040.81 \pm 457.60$ | $1030.48 \pm 90.27$ | $1 \times 10^{-6} \pm 3 \times 10^{-7}$ | $0.5657 \pm 0.0501$ | $0 \pm 0$ |
| DFR-SGL | SCHEETZ | 68084.77 | 2246.13 | 30.31 | $1891.21 \pm 386.40$ | $642.38 \pm 136.12$ | $3 \times 10^{-9} \pm 8 \times 10^{-10}$ | $0.1818 \pm 0.0390$ | $0 \pm 0$ |
| SPARSEGL | SCHEETZ | 68084.77 | 6666.65 | 10.21 | $1891.21 \pm 386.40$ | $1890.45 \pm 386.44$ | $5 \times 10^{-22} \pm 2 \times 10^{-22}$ | $0.1818 \pm 0.0390$ | $0.1818 \pm 0.0390$ |
| DFR-ASGL | TRUST-EXPERTS | 5.96 | 3.10 | 1.92 | $76.74 \pm 1.80$ | $85.11 \pm 3.96$ | $1 \times 10^{-11} \pm 3 \times 10^{-12}$ | $0 \pm 0$ | $0 \pm 0$ |
| DFR-SGL | TRUST-EXPERTS | 7.29 | 3.20 | 2.28 | $98.36 \pm 4.17$ | $92.55 \pm 5.93$ | $4 \times 10^{-11} \pm 9 \times 10^{-12}$ | $0 \pm 0$ | $0 \pm 0$ |
| SPARSEGL | TRUST-EXPERTS | 7.29 | 4.52 | 1.61 | $98.36 \pm 4.17$ | $104.24 \pm 5.15$ | $2 \times 10^{-6} \pm 2 \times 10^{-6}$ | $0 \pm 0$ | $0 \pm 0$ |
| DFR-ASGL | TUMOUR | 7027.69 | 86.25 | 81.48 | $8783.1 \pm 265.36$ | $82.75 \pm 5.61$ | $3 \times 10^{-8} \pm 9 \times 10^{-9}$ | $0.6768 \pm 0.0472$ | $0 \pm 0$ |
| DFR-SGL | TUMOUR | 7466.83 | 89.43 | 83.49 | $9272.02 \pm 224.60$ | $186.27 \pm 7.64$ | $3 \times 10^{-9} \pm 2 \times 10^{-10}$ | $0.8586 \pm 0.0352$ | $0 \pm 0$ |
| SPARSEGL | TUMOUR | 7466.83 | 90.03 | 82.93 | $9272.02 \pm 224.60$ | $197.40 \pm 9.38$ | $2 \times 10^{-9} \pm 2 \times 10^{-10}$ | $0.8586 \pm 0.0352$ | $0 \pm 0$ |

