# OpenReview forum: "Dual Feature Reduction for the Sparse-group Lasso and its Adaptive Variant"
_ICML.cc/2025/Conference — ICML 2025 poster_

### Official Review · Reviewer_mWUc · 2025-03-12

**Overall Recommendation:** 3

**Summary:**

The #6131 presents dual feature reduction framework, a novel bilevel screening method specifically for the sparse-group Lasso (SGL) and its adaptive variant (aSGL). SGL works by applying $\ell\_1$ (variable-level) and $\ell_2$ (group-level) shrinkage, and the paper's problem setting minimizes a convex differentiable loss function $f(\beta)$ with a sparse-group penalty $\\|\beta\\|\_{\mathrm{sgl}}=\alpha\\|\beta\\|\_1+(1-\alpha) \sum\_{g=1}^m \sqrt{p\_g}\left\\|\beta^{(g)}\right\\|\_2$. The proposed dual reduction  technique reduces computational complexity by pre-screening and eliminating inactive groups and variables before it runs into optimization. It achieves this using strong screening rules derived from dual norms and Lipschitz assumptions on the gradients, specifically leveraging the subdifferential characterizations of the SGL norm. DFR performs bi-level screening, i.e., initially at the group-level (discarding groups satisfying $\left\\|\nabla\_g f\left(\hat{\beta}\left(\lambda\_k\right)\right)\right\\|\_{\epsilon\_g} \leq \tau\_g\left(2 \lambda\_{k+1}-\lambda\_k\right)$ ) and then variable-level within active groups (discarding variables when $\left|\nabla\_i f\left(\hat{\beta}\left(\lambda\_k\right)\right)\right| \leq$ $\alpha\left(2 \lambda\_{k+1}-\lambda\_k\right)$ ). KKT conditions are checked to correct any screening violations, ensuring optimality.

Lastly, their numerical experiments (synthetic and real datasets) show the DFR significantly reduces computational cost while maintaining robustness and achieving identical optimal solutions to standard SGL methods.

**Claims And Evidence:**

The evidence is generally convincing.

**Essential References Not Discussed:**

There do not appear to be any significant missing references.

**Experimental Designs Or Analyses:**

They are reasonable from my perspective.

**Methods And Evaluation Criteria:**

Yes the evaluation criteria are appropriate. The work uses metrics like the improvement factor (the ratio of computational time with and without screening) and input proportion (the fraction of variables retained after screening), which directly measure the efficiency gains from the proposed method.

**Other Comments Or Suggestions:**

Integrating the pseudocode from the appendix into the main body could improve readability and help readers grasp the key ideas more effectively.

**Other Strengths And Weaknesses:**

- Although KKT checks prevent incorrect feature elimination, they introduce additional computational costs that are not explicitly benchmarked in isolation. The paper notes increased KKT violations for DFR-aSGL compared to DFR-SGL, suggesting the Lipschitz assumptions are less robust when adaptive penalties are introduced.

- While DFR is empirically faster, the computation of the $\epsilon$-norm has a worst-case complexity of $O(p_g log p_g)$, which could become a bottleneck for very large group sizes.

**Questions For Authors:**

1. Can your DFR be extended to handle nonlinear models like kernel-based methods or neural networks, or other losses beyond simple square and logistic loss? If not, what are the fundamental barriers to applying this method beyond convex optimization problems?

2. The approach is developed specifically for convex SGL penalties, and it's unclear if it would extend to non-convex sparse-group penalties like sparse-group SCAD.

**Relation To Broader Scientific Literature:**

The paper’s key contributions build on a rich literature in sparse estimation and feature screening rule.  Prior work on screening rules for the lasso (most notably the strong rules by Tibshirani et al. 2010) and safe screening techniques (like those by El Ghaoui et al. 2010) provided the conceptual and mathematical framework for discarding inactive features prior to optimization.

**Theoretical Claims:**

I did not find any major mathematical mistakes.

The proof (as with many strong rules) relies on a Lipschitz assumption for the gradients. In practice, if the loss function $f$ does not have a Lipschitz‐continuous gradient (or, if the constant is underestimated), the assumption might fail. The authors are aware of this issue and use KKT checks to guard against potential violations. The derivation of the KKT check conditions (expressed in terms of a soft-thresholding operator) as well as many tools is standard in literature series.

---

> ### Author Rebuttal · Authors · 2025-03-27
>
> We thank the reviewer for the time and thought they invested in our manuscript and in providing helpful feedback. In the camera-ready version, we will add the pseudocode from the appendix into the main text to improve readability. In response to specific points raised:
>
> >Although KKT checks prevent incorrect feature elimination, they introduce additional computational costs that are not explicitly benchmarked in isolation. The paper notes increased KKT violations for DFR-aSGL compared to DFR-SGL, suggesting the Lipschitz assumptions are less robust when adaptive penalties are introduced.
>
> We did not provide a detailed timing breakdown as **our focus was on end-to-end efficiency for model users**, which is most relevant in practice. We agree that a detailed runtime breakdown would be insightful. We have added an analysis below, and we will add a more comprehensive analysis with figures in the camera-ready version.
>
> *Breakdown comparison*: We ran a breakdown analysis for Figure 1 and found that on average across all values of $\alpha$ the following breakdown occurred for fitting a full path (as a percentage of the total runtime):
>
> * **Fitting algorithm.** DFR-SGL: 88% ($133$s), DFR-aSGL: 86% ($134$s).
> * **$\epsilon$-norm evaluation.** DFR-SGL: 3.9% ($3$s), DFR-aSGL: 3.6% ($3$s).
> * **Group screening.** DFR-SGL: 3.9% ($3$s), DFR-aSGL: 3.6% ($3$s).
> * **Variable screening.** DFR-SGL: 0.01% ($0.01$s), DFR-aSGL: 0.01% ($0.01$s).
> * **KKT checks.** DFR-SGL: 0.6% ($0.4$s), DFR-aSGL: 0.6% ($0.4$s).
>
>  Additionally, we ran a breakdown analysis on the real dataset *scheetz* for fitting a full path:
> * **Fitting algorithm.** DFR-SGL: 77% ($275$s), DFR-aSGL: 65% ($244$s).
> * **$\epsilon$-norm evaluation.** DFR-SGL: 0.46% ($1.64$s), DFR-aSGL: 0.57% ($2$s).
> * **Group screening.** DFR-SGL: 0.46% ($1.66$s), DFR-aSGL: 0.58% ($2.18$s).
> * **Variable screening.** DFR-SGL: 0.01% ($0.01$s), DFR-aSGL: 0.02% ($0.08$s).
> * **KKT checks.** DFR-SGL: 0.2% ($0.68$s), DFR-aSGL: 0.3% ($1.14$s).
>
> The analysis shows that screening adds minimal overhead while significantly improving fitting efficiency. In Figure 1, without screening, the average fitting time was $544$s for SGL and $603$s for aSGL.
>
> > Can your DFR be extended to handle nonlinear models like kernel-based methods or neural networks, or other losses beyond simple square and logistic loss? If not, what are the fundamental barriers to applying this method beyond convex optimization problems?
>
> The core assumptions are that the **loss function is convex and differentiable**, so DFR can be applied to any loss function satisfying these. Convexity prevents multiple optimal solutions that could complicate screening, and we require differentiability to derive the screening rules and KKT checks (we need access to $\nabla f$).
>
> In the manuscript, we have focused on linear and logistic regression as the solver used, ATOS, has the additional assumption that the loss must also have a Lipschitz gradient (also known as L$-smooth). However, we have also showcased that DFR can be efficiently implemented using BCD, which does not have these limitations. Therefore, a potential future direction can be the exploration of applying DFR to other loss functions.
>
> >The approach is developed specifically for convex SGL penalties, and it's unclear if it would extend to non-convex sparse-group penalties like sparse-group SCAD.
>
> DFR, like any screening rule, is derived using the subdifferentials of SGL. Extending it to other penalties requires deriving their subdifferentials. Strong rules work best with uniqueness properties on subgradients, well-behaved KKT conditions, and strong duality—properties that are often absent in non-convex cases. Thus, DFR is not inherently limited to SGL, aside from the usual challenges of screening non-convex penalties, as discussed.
>
> The two-layer screening framework used for DFR applies to any sparse-group model, but literature on screening rules applied to non-convex penalties is relatively light (see (a)).
>
> **Please don't hesitate to ask us any additional questions about our work.**
>
> *References*
>
> (a) Alain Rakotomamonjy, et al. “Screening rules for Lasso with non-convex Sparse Regularizers”. PMLR, 2019.

---

> > ### Comment · Reviewer_mWUc · 2025-04-05
> >
> > Thank you for providing additional simulations. I reviewed this work on previous conference and based on the rebuttal I don't have other concerns.

---

### Official Review · Reviewer_yM6R · 2025-03-13

**Overall Recommendation:** 3

**Summary:**

This paper introduces Dual Feature Reduction (DFR), a novel screening method to enhance the computational efficiency of Sparse-Group Lasso (SGL) and its adaptive variant (aSGL).

DFR applies two-layer screening:
- Group Reduction eliminates inactive groups using a strong screening rule based on dual norms and KKT conditions.
- Variable Reduction further removes inactive features within active groups.
DFR is the first bi-level strong screening method for SGL and the first screening rule for aSGL, producing the same optimal solution with significantly lower computational cost than GAP Safe and sparsegl.

Experiments on synthetic and real-world datasets show that DFR achieves significant speedup while maintaining selection accuracy. It enables expanded hyperparameter tuning and makes SGL more scalable for high-dimensional learning tasks, particularly in ML modeling in genetics.

**Claims And Evidence:**

**Well-Supported Claims**
1. DFR reduces computational cost while preserving solution optimality.
   - Evidence: The theoretical analysis shows that DFR maintains the same optimal solution by leveraging dual norms and KKT-based strong screening rules.
   - Experimental Support: DFR substantially lowers the number of variables to be optimized, resulting in significant computational savings.
2. DFR outperforms existing screening methods.
   - Evidence: Experiments indicate that DFR reduces the input dimensionality more effectively than both GAP and sparsegl, yielding a lower optimization cost.
3. DFR enables expanded hyperparameter tuning.
   - Evidence: Because DFR significantly cuts computation time, it becomes feasible to jointly tune \(\lambda\) and \(\alpha\) in SGL-based models.

**Claims Requiring More Evidence**
1. The “Improvement Factor” (IF) is a fair and reliable efficiency metric.
   - Issue: While IF appears to be based on overall runtime, the paper does not explicitly confirm this. Its fairness is uncertain for several reasons:
     - Baseline Dependence: If the no-screening solver is suboptimal, IF might overstate DFR’s efficiency gain.
     - KKT Overhead: Without a runtime breakdown for screening, KKT checks, and optimization, it’s unclear whether KKT overhead negates the benefit of reduced dimensionality.
     - Comparison with GAP Safe: IF could favor heuristic methods like DFR over exact approaches like GAP Safe, which might be more conservative in screening.
2. DFR outperforms GAP Safe.
   - Issue: The paper does not compare GAP Safe in certain synthetic scenarios (e.g., high dimensionality, uneven group sizes, logistic models) and omits real-data comparisons. Therefore, the practical advantage of DFR over GAP Safe remains uncertain.
3. DFR’s two-layer screening does not significantly increase false exclusions.
   - Issue: There were some KKT violations, especially in the adaptive SGL setting. The frequency and consequences of these violations are not thoroughly discussed, leaving open questions about screening accuracy.

**Essential References Not Discussed:**

NA

**Experimental Designs Or Analyses:**

**Strengths**
1. Broad Experimental Scope: The paper evaluates DFR on synthetic datasets (varying sparsity, correlation, dimensionality) and real datasets (genetics, classification), capturing both controlled and practical scenarios.
2. Comparison with Established Baselines: In parts of the synthetic experiments, GAP Safe and sparsegl are used as benchmarks, offering a relevant performance context for DFR.
3. Robustness Checks: The analyses investigate factors like signal strength, data correlation, and logistic vs. linear models, highlighting how DFR fares under diverse conditions.

**Potential Issues**
1. Incomplete Details for Increasing Dimensionality: While DFR is tested in high dimensions, the paper lacks a clear description of how the synthetic data are generated in the increasing-dimensionality scenarios. This hampers reproducibility and leaves open questions about the underlying correlation structures and signal placement.
2. Omission of GAP Safe in Later Synthetic Tests: In the high dimensionality, robustness and logistic experiments, GAP Safe is not included despite being considered earlier. This limits insight into how GAP Safe compares to DFR under these more diverse conditions.
3. Reliance on Improvement Factor: Though informative, the improvement factor might overlook the runtime overhead for KKT checks and solver differences. Absolute runtime or time-breakdown analyses could provide a fuller picture.
4. Solver Consistency: The paper does not clearly state if all methods (DFR, GAP Safe, sparsegl) use identical solver settings, raising potential fairness concerns in runtime comparisons.

**Methods And Evaluation Criteria:**

**Strengths**
1. DFR directly addresses SGL’s computational bottleneck, leveraging dual norms and KKT-based screening for feature reduction.
2. Comprehensive evaluation includes synthetic data (controlled sparsity, correlation, dimensionality) and real datasets (genetics, machine learning).
3. Baseline comparisons with GAP Safe (exact screening) and sparsegl (heuristic screening) contextualize DFR’s performance.

**Potential Issues**
1. GAP Safe is omitted from several synthetic data and real data experiments, leaving uncertainty about its practical efficiency.
2. Computational breakdown is missing, making it unclear how much speedup comes from screening vs. solving the reduced problem. Please consider provide absolute runtime comparisons and a screening vs. KKT vs. optimization time breakdown.
3. Need to clarify solver settings across all methods. Potential solver bias if different methods are not optimized consistently.

**Other Comments Or Suggestions:**

- Reorganize Experimental Section: The paper’s experiment write-up often references both the main text and multiple Appendix sections, causing fragmented reading. Consider consolidating key experimental details or providing clearer cross-references so readers don’t have to jump back and forth.
- Clarify Improvement Factor Usage: Although the paper uses the improvement factor metric, it might overlook runtime overhead (e.g., KKT checks). Presenting absolute runtimes or a time breakdown alongside the improvement factor would help address fairness concerns.

**Other Strengths And Weaknesses:**

NA

**Questions For Authors:**

1. The paper does not compare GAP Safe in certain synthetic scenarios (e.g., increased dimensionality, uneven group sizes, logistic models) and omits it for real data. Could you provide results—or at least partial findings—on those settings to clarify whether DFR consistently outperforms GAP Safe across diverse conditions?
2. Would it be possible to offer a more detailed runtime breakdown—including screening, KKT checks, and reduced optimization—alongside the improvement factor?
3. Could you elaborate on how the synthetic datasets were generated for the increasing-dimensionality experiments (e.g., correlation structures, signal placement, group definitions)?

**Relation To Broader Scientific Literature:**

This paper extends strong screening frameworks (Tibshirani et al., 2010) to the Sparse-Group Lasso (SGL) and adaptive SGL, building on prior safe/strong rules for the group lasso (e.g., Ndiaye et al., 2016). By employing dual norms and KKT-based subgradients, the method follows the dual polytope projection logic found in other screening approaches (Wang et al., 2013). However, unlike GAP Safe (Ndiaye et al., 2016) which iterates for exact screening, DFR uses single-pass strong rules coupled with KKT checks. The bi-level screening structure aligns with earlier group-sparse penalties but adds a second screening stage to reduce dimensionality within each group—an innovation that refines prior one-layer methods like sparsegl (Liang et al., 2022). By extending the framework to adaptive SGL, the paper addresses the oracle property aspect (Poignard, 2020), contributing to literature on sparsity-inducing regularization with adaptive weights.

**Theoretical Claims:**

Generally checked and no major issues found in the submission.

---

> ### Author Rebuttal · Authors · 2025-03-27
>
> We want to thank the reviewer for taking the time to review our work and for their helpful comments. In the camera-ready version, the experimental section will be restructured to improve readability, reducing the need for frequent cross-referencing. In response to specific points raised:
>
> >Claims Requiring More Evidence: The “Improvement Factor” (IF) is a fair and reliable efficiency metric.
>
> We use IF as it provides  **direct insight into how screening impacts the user**. For instance, an IF of $2$ means DFR halves fitting time. The metric does not inherently favour any method. We also report Input Proportion as an alternative screening impact measure, independent of computational considerations. Both metrics show DFR is effective. Additionally, raw runtimes are available in the appendix, enabling direct method comparison.
>
> >If the no-screening solver is suboptimal, IF might overstate DFR’s efficiency gain.
>
> We considered this when using ATOS with DFR. To ensure IF does not unfairly benefit DFR over GAP safe due to the solver, we also implemented DFR with BCD (the solver used for GAP) and found similar results (Figure 1).
>
> >Claims Requiring More Evidence: DFR’s two-layer screening does not significantly increase false exclusions.
>
> We detailed our findings on KKT violations in the 'KKT violations' section (Lines 363-377). The higher violations in aSGL likely stem from the Lipschitz assumptions' dependence on additional hyperparameters. However, across all results (synthetic and real), KKT violations for DFR-SGL and DFR-aSGL were minimal. Specifically, DFR-SGL had only one violation overall, and DFR-aSGL had a violation every $108$ fits (Figures 1-3, Table A4).
>
> >Would it be possible to offer a more detailed runtime breakdown—including screening, KKT checks, and reduced optimization—alongside the improvement factor?
>
> We did not provide a detailed timing breakdown as our focus was on end-to-end efficiency for model users, which is most relevant in practice. We agree that a detailed runtime breakdown would be insightful. We have added an analysis below, and we will add a more comprehensive analysis with figures in the camera-ready version.
>
> *Breakdown comparison*: We ran a breakdown analysis for Figure 1 and found that on average across all values of $\alpha$ the following breakdown occurred for fitting a full path (as a percentage of the total runtime):
>
> * **Fitting algorithm.** DFR-SGL: 88% ($133$s), DFR-aSGL: 86% ($134$s).
> * **$\epsilon$-norm evaluation.** DFR-SGL: 3.9% ($3$s), DFR-aSGL: 3.6% ($3$s).
> * **Group screening.** DFR-SGL: 3.9% ($3$s), DFR-aSGL: 3.6% ($3$s).
> * **Variable screening.** DFR-SGL: 0.01% ($0.01$s), DFR-aSGL: 0.01% ($0.01$s).
> * **KKT checks.** DFR-SGL: 0.6% ($0.4$s), DFR-aSGL: 0.6% ($0.4$s).
>
>  Additionally, we ran a breakdown analysis on the real dataset *scheetz* for fitting a full path:
> * **Fitting algorithm.** DFR-SGL: 77% ($275$s), DFR-aSGL: 65% ($244$s).
> * **$\epsilon$-norm evaluation.** DFR-SGL: 0.46% ($1.64$s), DFR-aSGL: 0.57% ($2$s).
> * **Group screening.** DFR-SGL: 0.46% ($1.66$s), DFR-aSGL: 0.58% ($2.18$s).
> * **Variable screening.** DFR-SGL: 0.01% ($0.01$s), DFR-aSGL: 0.02% ($0.08$s).
> * **KKT checks.** DFR-SGL: 0.2% ($0.68$s), DFR-aSGL: 0.3% ($1.14$s).
>
> The analysis shows that screening adds minimal overhead while significantly improving fitting efficiency. In Figure 1, without screening, the average fitting time was $544$s for SGL and $603$s for aSGL.
>
> >Need to clarify solver settings across all methods. Potential solver bias if different methods are not optimized consistently.
>
> See lines 260-266 and the 'Comparison to BCD' section, where we state that DFR uses ATOS while GAP safe uses BCD. To check the solver does not bias results, we did implement DFR with BCD and found similar outcomes (Figure 1).
>
> >The paper does not compare GAP Safe in certain synthetic scenarios (e.g., increased dimensionality, uneven group sizes, logistic models) and omits it for real data.
>
> The GAP safe rules failed in most simulations we tested, with issues in convergence and solution optimality. Using the authors' implementation (see Link 1), we are confident the problem lies with the approach, not the code. We appreciate that a comparison is important, so we included it where feasible (Figures 1-3). The poor performance suggested further investigation would not be fruitful.
>
> Link 1: https://github.com/EugeneNdiaye/Gap\_Safe\_Rules
>
> >Could you elaborate on how the synthetic datasets were generated for the increasing-dimensionality experiments (e.g., correlation structures, signal placement, group definitions)?
>
> For the increasing dimension case, the setting is the same as for the other cases (described in Section 3.1), aside from the grouping structure. As explained in the 'Increasing dimensionality' section, the variables were grouped into groups of sizes $20$, so that there were $p/20$ groups for each case of $p$.
>
> **Please don't hesitate to ask us any additional questions about our work.**

---

### Official Review · Reviewer_fLQH · 2025-03-21

**Overall Recommendation:** 4

**Summary:**

This paper introduces a new feature reduction method in order to improve the computational complexity in solving Sparse-Group Lasso (SGL) problems.
The Dual Feature Reduction (DFR) method that is presented relies on two screening stages (one for inactive groups and another for inactive variables within a group) and the authors provide both theoretical groundings and experimental results to support their claims.

### Update after rebuttal.
I have acknowledged the response of the authors and will maintain my recommendation. Although for the completeness of the experimental protocol, I would still recommend including GAP Safe in all experiments.

**Claims And Evidence:**

The authors provide evidence through coherent theoretical bases building upon well-established literature, and convincing empirical results.
Refer to §Methods and evaluation criteria for more details.

**Essential References Not Discussed:**

N.A.

**Experimental Designs Or Analyses:**

I have checked the soundness of the experimental design (both synthetic and real data). Please refer to §Questions.

**Methods And Evaluation Criteria:**

The problem at hand is concerned with reducing the computational costs in solving SGL problems.
It is well established in the screening literature that reducing the input feature size by identifying those that will be inactive at the optimal solution induces significant computational gains to the subsequent optimization scheme.
The proposed DFR method does therefore make sense for the problem at hand.

In their experimental evaluations, the authors clearly distinguish the gain in performance through what they call the _improvement factor_ (the ratio of computation time in methods without and with screening approach) and the _input proportion_ (quantifying how much the feature space was reduced). Their proposed DFR method is compared against the sparsegl and GAP Safe approaches, which are two other competing rules of screening for SGL problems.

**Other Comments Or Suggestions:**

For improved readability, I would suggest increasing the font sizes of axis labels in all the figures.

**Other Strengths And Weaknesses:**

To the best of my knowledge, the DFR method proposed by the authors is novel.
The presentation of their approach is clear and gradual, building upon well-established existing literature.
Their experimental evaluation is thorough (though please also refer to §Questions).
I believe this is a valid contribution to the broader scientific community.

**Questions For Authors:**

Why is the GAP Safe approach not included in the evaluations of the impact of dimensionality, sparsity proportion or data correlation (Figs 4 to 6), or in the real data analysis (Figs 8 and 9) ?

**Relation To Broader Scientific Literature:**

The key contributions of this paper are regarding screening rules for Sparse-Group Lasso problems, which are commonly found in various machine learning fields where relevant groups of variables need to be identified while ensuring limited spurious variables within a given group.

**Theoretical Claims:**

I have not checked the proofs of the theoretical claims.

---

> ### Author Rebuttal · Authors · 2025-03-27
>
> We want to thank the reviewer for their time in reviewing our work and for their helpful and positive feedback. As suggested, we will increase the font sizes in the figures in the camera-ready version. With regards to your question:
>
> >Why is the GAP Safe approach not included in the evaluations of the impact of dimensionality, sparsity proportion or data correlation (Figs 4 to 6), or in the real data analysis (Figs 8 and 9)?
>
> We found that the GAP safe rules did not work for most simulation settings we tried. We encountered issues with convergence and solution optimality. We used an implementation provided by the authors of the GAP safe rules (see Link 1), so that we are confident the issue was not with the implementation but with the approach itself. We understand that a comparison to the safe rules is important, so we included the comparison when it was possible (Figures 1-3). The poor performance of GAP safe in these settings convinced us that further investigation of the safe rules would not be fruitful.
>
> Link 1: https://github.com/EugeneNdiaye/Gap\_Safe\_Rules
>
> **Please don't hesitate to ask us any additional questions about our work.**

---

### Decision · Program_Chairs · 2025-05-01

**Decision:**

Accept (poster)

**Comment:**

The paper proposes a new screening rule for the Sparse Group Lasso. As many screening rules, it relies on duality and KKT conditions, at both group leavel and the feature level.
Though this is already a fairly studied area of research, in some settings, the proposed approach seems to improve upon the bsaelines, Gap Safe Screening rules of Ndiaye et al (2016) and the more recent sparsegl.